

# Mapping snow depth in alpine terrain with unmanned aerial systems (UAS): potential and limitations

**Y. Bühler[1], M. S. Adams[2], R. Bösch[3], and A. Stoffel[1]**

[1]WSL Institute for Snow and Avalanche Research SLF, Davos, Switzerland
[2]Austrian Research Centre for Forests (BFW), Innsbruck, Austria
[3]Swiss Federal Institute for Forest, Snow and Landscape Research WSL, Birmensdorf, Switzerland

Received: 30 November 2015 – Accepted: 14 December 2015 – Published: 14 January 2016

Correspondence to: Y. Bühler (buehler@slf.ch)

Published by Copernicus Publications on behalf of the European Geosciences Union.

**TCD**

doi:10.5194/tc-2015-220

**Mapping snow depth in alpine terrain with unmanned aerial systems (UAS)**

Y. Bühler et al.



**TCD**

doi:10.5194/tc-2015-220

**Mapping snow depth in alpine terrain with unmanned aerial systems (UAS)**

Y. Bühler et al.

## Abstract

Detailed information on the spatiotemporal distribution, and variability of snow depth (HS) is a crucial input for numerous applications in hydrology, climatology, ecology and avalanche research. Nowadays, snow depth distribution is usually estimated by
5 combining point measurements from weather stations or observers in the field with spatial interpolation algorithms. However, even a dense measurement network is not able to capture the large spatial variability of snow depth present in alpine terrain.

Remote sensing methods, such as laser scanning or digital photogrammetry, have recently been successfully applied to map snow depth variability at local and regional
scales. However, such data acquisition is costly if manned airplanes are involved. The effectiveness of ground-based measurements on the other hand is often hindered by occlusions, due to the complex terrain or acute viewing angles. In this paper, we investigate the application of unmanned aerial systems (UAS), in combination with structure-from-motion photogrammetry, to map snow depth distribution. Such systems have the
advantage that they are comparatively cost-effective and can be applied very flexibly to cover otherwise inaccessible terrain. In this study, we map snow depth at two different locations: (a) a sheltered location at the bottom of the Flüela valley (1900 m a.s.l.) and (b) an exposed location on a peak (2500 m a.s.l.) in the ski resort Jakobshorn, both in the vicinity of Davos, Switzerland. At the first test site, we monitor the ablation
on three different dates. We validate the photogrammetric snow depth maps using simultaneously acquired manual snow depth measurements. The resulting snow depth values have a root mean square error (RMSE) better than 0.07 to 0.15 m on meadows and rocks and a RMSE better than 0.30 m on sections covered by bushes or tall grass. This new measurement technology opens the door for efficient, flexible, repeatable and
cost effective snow depth monitoring for various applications, investigating the worlds cryosphere.

Discussion Paper | Discussion Paper | Discussion Paper | Discussion Paper | Discussion Paper |

**TCD**

doi:10.5194/tc-2015-220

**Mapping snow depth in alpine terrain with unmanned aerial systems (UAS)**

Y. Bühler et al.

# 1 Introduction

Information on the spatiotemporal distribution of snow depth is important for numerous applications: As it is a robust indicator for the amount of water stored as snow (snow water equivalent – SWE) (Jonas et al., 2009), it has a substantial impact on water sup-
5 ply and hydropower; the quality of hazard forecasting for floods and snow avalanches depends substantially on snow depth information (Bavay et al., 2009; McClung and Schaerer, 2006); the growth and habitat patterns of alpine flora and fauna is linked to the seasonal snow depth distribution (Bilodeau et al., 2013; Mysterud et al., 2001; Wipf et al., 2009); annual changes in snow depth over the winter season have strong impact
on alpine tourism as more and more ski resorts depend on technical snow production.

Numerous studies report a very high spatial variability of snow depth within small distances, in particular in alpine terrain (Egli et al., 2011; Elder et al., 1998; Grünewald et al., 2010; Schweizer et al., 2008). Remote sensing is a promising tool to monitor this spatial variability, because it can provide spatially continuous measurements at a
15 high spatial resolution of otherwise inaccessible areas. We define snow depth (HS) according to Fierz et al. (2009) as the vertical distance from the base to the snow pack surface at a specific location.

Terrestrial laser scanning (TLS) has been successfully applied in many case studies to measure HS distribution in small catchments with high vertical accuracies in the
20 range of 0.10 m (Deems et al., 2013; Grünewald et al., 2010; Melvold and Skaugen, 2013; Mott et al., 2010; Prokop, 2008; Schaffhauser et al., 2008). A recent study by Deems et al. (2015) uses TLS to visualize the HS distribution in avalanche release zones for the education of ski resort staff and assesses the different error sources. However, TLS-accuracies suffer from acute illumination angles, resulting in unfavor-
25 able laser footprints, in particular within flat areas. Furthermore, terrain sections behind convex landforms such as hills or moraines cannot be covered. Airborne laser scanning (ALS) on the other hand is still very costly (e.g. Bühler et al., 2015a). Therefore,

Discussion Paper | Discussion Paper | Discussion Paper | Discussion Paper |

digital photogrammetry is a promising and economic option for HS mapping in alpine terrain, in particular if it can be performed with cost-effective UAS.

First attempts to map snow depth with photogrammetry from manned aircrafts were already made decades ago (Cline, 1994, 1993; Smith et al., 1967). However the reported efficiency and the achieved accuracies of more than one meter were not feasible to most applications. With the advent of digital photogrammetry, this changed fundamentally. Recent investigations report accuracies in the range of centimeters to decimeters, which allow a detailed analysis of the spatial variability of the mountain snow cover (Bühler et al., 2015a; Lee et al., 2008; Nolan et al., 2015) but still require a fully equipped manned aircraft and corresponding maintenance logistics.

Throughout the last years, UAS have been used for a wide range of mapping and monitoring studies in mountainous regions, especially with a focus on natural hazards: Fernández et al. (2015) provide an extensive overview of recent surveys of landslides; Ryan et al. (2015) and Whitehead et al. (2013) reported on UAS applications on glaciers; Danzi et al. (2013) for rockfall, Dall'Asta et al. (2015) for rock glacier and Tampubolon and Reinhardt (2015) on volcano mapping. Enßle et al. (2015) successfully tested UAS-data acquisition in elevations up to 4200 m a.s.l., proving that UAS are capable of operating even at very high altitudes. However, to this date, the number of studies dealing with UAS-based photogrammetry to map snow and avalanche are very limited: First results have recently been published by De Michele et al. (2015), Eckerstorfer et al. (2016) and Jagt et al. (2015). Additionally, Basnet et al. (2015), Prokop et al. (2015) and Thibert et al. (2015) reported on using ground-based photogrammetry for snow and avalanche detection. De Michele et al. (2015) conclude, that UAS-based HS mapping holds great potential, but that further studies are required especially with regard to multi-temporal mapping, implementing sensors capable of measuring at e.g. near infrared wavelengths, or mapping different snow cover conditions or topographic areas.

**TCD**

doi:10.5194/tc-2015-220

**Mapping snow depth in alpine terrain with unmanned aerial systems (UAS)**

Y. Bühler et al.

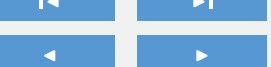

Discussion Paper | Discussion Paper | Discussion Paper | Discussion Paper

**TCD**

doi:10.5194/tc-2015-220

**Mapping snow depth in alpine terrain with unmanned aerial systems (UAS)**

Y. Bühler et al.

## 2 Methods: UAS and data processing

### 2.1 UAS AscTec Falcon 8

The UAS missions have been performed with an Ascending Technologies (AscTec) Falcon 8 octocopter equipped with a customized Sony NEX-7 camera. The Falcon 8 has been in serial production since 2009 and can be customized with different sensor systems. The system weighs 2.3 kg (incl. camera) and can be transported to remote locations fully assembled in a special backpack, a prerequisite for most alpine applications. A combination of onboard navigation sensors (Global Navigation Satellite System GNSS, Inertial Measurement Unit IMU, barometer and compass) and an adaptive control unit, permit high positional accuracy and stable flight characteristics even in challenging, alpine environmental conditions. The specifications of the Falcon 8 are listed in Table 1.

### 2.2 Technical specifications of the Falcon 8 UAS

The Sony NEX-7 system camera features a 24 MP APS-C CMOS sensor and is equipped with a small and lightweight Sony NEX 20 mm F/2.8 optical lens (81 g). By removing the built-in near infrared filter, the camera sensor is also sensitive above the red spectrum. This allows us to mount the lens with different filters such as visible colors (RGB) and near infrared (NIR) bands ($\lambda > 550$ nm, $\lambda > 770$ nm and $\lambda > 830$ nm) and without filter the camera sensor operates in a combined visual and NIR range. The near infrared sensitivity has advantages for snow (Bühler et al., 2015b) and vegetation (Tucker, 1979) analysis. The camera is connected to the Falcon 8 by a gimbal with active stabilization and vibration damping and is powered by the UAS battery. The viewfinder of the camera is transmitted to the ground control station as video signal and the basic camera functions such as the exposure time can be controlled from the ground.

The UAS missions are planned using the AscTec Navigator software on a tablet computer. Topographic maps are imported and the waypoint navigation is calculated based on camera specifications, desired ground sampling distance (GSD) and image overlap. At the location of a planned mission the tablet computer is connected to the ground con-
5 trol station and last corrections to the flight plan, e.g. due to unexpected terrain variations, can be applied. During the flight mission, the UAS automatically moves from way-point to waypoint, only the launch and final landing phase require manual interaction. UAS must be operated within visual line of sight and the pilot has to be ready to interrupt the flight at any time, as requested by Swiss regulations (http://www.bazl.admin.ch).

Portability of the UAS, a high image resolution and the ability to take off and land from an exposed site are important key features for photogrammetric UAS missions within alpine, snow-covered terrain. The Falcon 8 offers a feasible compromise between flight endurance, payload and stability in most conditions. The spectral and spatial capabilities of the Sony NEX-7 camera enable the generation of highly accurate digital
surface model (DSM). The portability is excellent as UAS, radio modem and controlling computer fit to a daypack. The short flight time per battery on the other hand, is a critical disadvantage of the octocopter technology. Longer flight times are the major advantage of fixed-wing UAS like the eBee (sensefly). However, their available cameras have only limited image resolution due to strict weight limits. Larger fixed-wing drones
like Sirius Pro from MAVinci, the UX5 from Trimble or the Q-200 from Quest UAS suffer from quite bulky overall equipment and are therefore not appropriate for high mountain areas. Feasible terrain (large flat areas) to safely land them does usually not exist. For an extensive overview of currently available UAS systems the reader is referred to Colomina and Molina (2014).

**2.3  Data processing**

The images are processed with Agisoft PhotoScan Pro v1.1.6, to generate georefer-enced DSMs and orthophotos. PhotoScan is based on a structure-from-motion (SfM) algorithm (Koenderink and van Doorn, 1991; Verhoeven, 2011) and implements a com-

**TCD**

doi:10.5194/tc-2015-220

**Mapping snow depth in alpine terrain with unmanned aerial systems (UAS)**

Y. Bühler et al.



**TCD**

doi:10.5194/tc-2015-220

**Mapping snow depth in alpine terrain with unmanned aerial systems (UAS)**

Y. Bühler et al.

plete photogrammetric workflow with special emphasis on multi-view reconstruction with UAS-based images. The tie point matching of PhotoScan allows the estimation of the internal and external camera orientation parameters and is followed by adding georeference information (coordinate system and ground control points). The result-ing model is linearly converted using the Helmert-Transform with 7 parameters and therefore compensates only for linear misalignment. Non-linear deformations from the model are removed by optimizing the estimated point cloud and camera parameters using 4 radial and 4 tangential distortion coefficients (Agisoft PhotoScan User Manual, http://www.agisoft.com/downloads/user-manuals/). During creation of the dense point cloud the estimated camera positions are used to calculate depth information for each camera and will be combined into a single dense point cloud. Two parameters of the dense cloud processing step have the strongest impact on the resulting point cloud:

1. "Quality" defines the desired reconstruction detail level. Higher quality settings can be used to obtain more detailed and more accurate geometry, but can results in significantly longer time for processing.

2. "Depth filtering" allows removing outliers from the point cloud, which are caused by poor texture of the scene, noisy or blurry images. Depending on the complexity of the scene geometry, different depth filtering modes can be applied. The accu-racy of the exported product needs to be analyzed to estimate the complexity in the model and thus select an appropriate depth-filtering mode.

## 3 Test sites and data acquisition

To test the feasibility of UAS-based HS change mapping, two well-accessible test sites in the region of Davos, Switzerland have been chosen and represent typical locations for HS studies in high alpine environment (Fig. 1).

Discussion Paper | Discussion Paper | Discussion Paper | Discussion Paper | Discussion Paper |

**TCD**

doi:10.5194/tc-2015-220

**Mapping snow depth in alpine terrain with unmanned aerial systems (UAS)**

Y. Bühler et al.

## 3.1 Tschuggen: sheltered valley bottom

The test site Tschuggen is at the bottom of the Flüela valley at an elevation of 1940 m a.s.l. very close to the timberline. This spot is well accessible even during the winter season, because the Flüela pass road is regularly cleared until this point. The high alpine valley bottom features both, quite flat alpine meadows and hilly alpine terrain. The main land cover is a mixture of bushes (mainly alpine rose, juniper and erica) containing steep rocky outcrops and sparse larch and pine trees (Fig. 10). Only moderate HS variability can be expected at this site in an average winter season because it is usually not exposed to high winds. The mean slope angle of the test site is 19° ranging from 0 to 80°. The reference measurement plots have been acquired in areas between 4 and 36° slope angle with an average slope angle of 20°.

A total of 252 images at 4 different dates have been acquired at this test site between March and September 2015 (Table 2; Fig. 2). An image overlap of 70 % along-track and across-track is a good compromise between the time required for data acquisition and quality of the resulting DSM. The first three flights were done with an old version of the AscTec flight control hardware, which required the UAS to stop and stabilize for every image acquisition, consuming considerably more time and energy to cover a specific area. The last data acquisition was performed with an updated software version where the UAS does not stop, enabling the acquisition of up to five times more images with one battery.

For the absolute orientation, selected ground control points (GCPs) have been selected, which were required to be clearly visible in the base imagery of all four acquisition dates. The GCPs, bright quartz marks on rocks and center lines of the road, have been measured with a Leica TPS 1200 differential GNSS with an expected accuracy of better than 0.03 m. The achieved average accuracy of the orthorectification process is 0.038 m ($x = 0.029$ m, $y = 0.021$ m, $z = 0.012$ m).

Simultaneously to the UAS data acquisition, HS reference measurements were acquired with a marked avalanche probe (Fig. 2). Five manual, plumb vertical measure-

**TCD**

doi:10.5194/tc-2015-220

**Mapping snow depth in alpine terrain with unmanned aerial systems (UAS)**

Y. Bühler et al.

ments within one square meter (at all corners and the center) have been carried out and the center point have been recorded with a Trimble GeoXH differential GNSS device with an expected accuracy better than 0.10 m.

## 3.2 Brämabühl: exposed mountain top

The test site Brämabühl is located at the top of the ski area Jakobshorn in Davos, Switzerland at an elevation of 2500 m a.s.l. and is approximately 5.5 km linear distance from the test site Tschuggen (Fig. 1). At this test site we expect a much higher variability of HS and in particular higher maximum HS values compared to the test site Tschuggen. The high wind exposure around the top of a crest at high elevation is expected to lead to a large amount of windblown snow. Additionally, the ski runs present within the area are typically areas for snow grooming and artificial snow production. The top of Brämabühl is covered mainly by high alpine meadow and small bushes (Fig. 10). No trees or larger bushes grow at this elevation and local climate. The mean slope angle of the test site is 30° ranging from 0° up to 90° in the small rock faces. The reference measurement plots have ben acquired at slope angles between 5 and 41° with a mean slope angle of 20°.

For this test site near infrared imagery has been selected, which is expected to have higher contrast and lower reflection on snow-covered areas (Bühler et al., 2015b). Table 3 shows the data acquisition information and Fig. 3 the resulting orthophotos, with a spatial resolution of 0.025 m. The same image overlap of 70 % along-track and cross-track, like at the Tschuggen test site, has been used. For the second field campaign, data acquisition was performed with the updated Falcon 8, explaining the much higher number of images and ground coverage in Table 3.

The image processing scheme from the Tschuggen experiment was repeated, but due to the smoother terrain with only a few clearly identifiable reference points, 10 artificial GCPs (white plastic sheets with a symmetric black cross in the middle) have been distributed and were measured with a Trimble GeoXH differential GNSS with an expected accuracy of better than 0.10 m. This approach allows a very accurate identi-

fication of the GCPs in the imagery. However, the distribution of the artificial GCPs is time consuming and a meaningful distribution over the test site is often not possible due to e.g. avalanche danger. In addition the applied Trimble GeoXH has a lower position-ing accuracy than the Leica TPS 1200 GNSS used at Tschuggen. Using 10 GCPs, the achieved referene accuracies of 0.019 m in $x$, 0.030 m in $y$ and 0.032 m in $z$ direction, are resulting in a combined error of 0.048 m.

The snow-covered imagery has been referenced by taking natural GCPs, which are clearly visible in the snow-free and snow-covered imagery (Fig. 3). The correspond-ing $x$, $y$ and $z$ coordinates of the snow-free imagery have been used to reference the snow-covered imagery. This approach ensures an accurate coregistration of the two DSMs. However, it is only possible if snow free areas contain enough well visible fea-tures that are sufficiently distributed over the test site. The achieved georeferencing accuracy with 10 control points is 0.155 m ($x = 0.079$ m, $y = 0.102$ m, $z = 0.086$ m), the result is worse than for the artificial GCPs, as the natural GCPs are harder to locate exactly.

Simultaneously to the winter UAS data acquisition, HS has been measured with a marked avalanche probe at 22 plots as reference data, locating the center points of the plots using the Trimble Geo XH GNSS.

## 4 Results and validation

### 4.1 Tschuggen: valley bottom

To produce the high spatial resolution (0.10 m) HS maps, the snow-free DSM (29 September 2015) has been subtracted from the snow-covered DSMs (11 March, 24 April and 12 May). These maps reveal the high spatial variability of HS already present at sheltered locations in alpine terrain (Fig. 4, top panels). Particularly in the southeastern part of the test site, areas with complex topography exist. Patches with nearly no snow in wind facing areas (luv) and pockets filled by windblown snow with

**TCD**

doi:10.5194/tc-2015-220

**Mapping snow depth in alpine terrain with unmanned aerial systems (UAS)**

Y. Bühler et al.

Discussion Paper | Discussion Paper | Discussion Paper | Discussion Paper

HS up to 2 m in the wind sheltered areas (lee) are connected within less than a meter distance. For the area depicted in Fig. 4, the mean HS $\overline{x}$ and the standard deviation $\sigma$ decrease from $\overline{x} = 0.66$ m and $\sigma = 0.36$ on 11 March to $\overline{x} = 0.31$ m and $\sigma = 0.31$ on 24 April and to $\overline{x} = 0.01$ m and $\sigma = 0.09$ on 12 May. Because of the produced HS maps from different dates, including approximately the peak of winter HS accumulation (11 March 2015), the spatial distribution of HS change as the percentage of remaining snow compared to the maximum HS can be calculated and visualized. Prior to the generation of the relative HS change maps the snow-covered areas have been classified using a simple unsupervised classification, based on the three spectral bands of the orthophoto. All areas not covered by snow have been set to zero HS. Isolated negative snow depth values, mainly caused by summer vegetation (Sect. 5.4), are not masked out but depicted as 0 HS in the maps.

The locations of the probe measurements are depicted in Fig. 2. We compare the mean $\overline{x}$ and the standard deviation $\sigma$ of the five manual measurements per plot with the $\overline{x}$ and $\sigma$ of all pixels ($10 \times 10 = 100$) within the 1 m$^2$ box around the center localized with differential GNSS. The results of this comparison are depicted in Fig. 5.

The HS root mean square error (RMSE) over all 50 reference plots is 0.25 m and there is an average systematic underestimation of HS by 0.2 m. For a more detailed analysis we divide the reference measurements in two classes based on the manual analysis of the 0.025 m spatial resolution snow-free orthophoto acquired on 28 September 2015: (a) *short grass/rocks* where no high vegetation is present and (b) *bushes/high grass*, where the surface of the dense vegetation is more than 0.10 m higher than the bare ground. In the second class the snow-free DSM is significantly higher than the terrain without vegetation. Because the snow presses the grass and bushes down to the ground in winter, the difference between the snow-covered and snow-free DSM results in a systematic underestimation of HS. For the class *short grass/rocks* the RMSE is 0.07 m and there is a mean shift of only 0.05 m. For the class *bushes/high grass* on the other hand the RMSE is 0.30 m and there is a mean shift of 0.29 m, corresponding to the mean height of bushes and tall grass within the investigation area. For snow

**TCD**

doi:10.5194/tc-2015-220

**Mapping snow depth in alpine terrain with unmanned aerial systems (UAS)**

Y. Bühler et al.

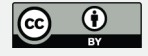

hydrological applications it is also important to gain information on the standard deviation $\sigma$ of HS within a specific plot. Even though the reference plots are only 1 m$^2$ we find $\sigma$ values up to 0.2 m. The RMSE of $\sigma$ is 0.04 m and there is no significant difference between the two investigated classes.

## 4.2 Brämabühl: mountain top

The HS map with a spatial resolution of 0.1 m shows different characteristics compared to the Tschuggen test site. The expected higher HS values of up to 5 m are clearly visible in Fig. 6. The close-up of the central part reveals interesting details such as the linear feature of buried hiking paths in the northwest or the snow grooming on the ski tracks. Over the entire area we calculate a mean HS $\overline{x} = 1.41$ m and $\sigma = 0.78$. Both $\overline{x}$ and $\sigma$ are more than twice as high as at the Tschuggen test site. The high spatial variability gets even more obvious in the 3-D view. We provide an animation of this 3-D visualization as Supplement to the paper (mp4 3-D movie). Snow filled bowls lay directly next to ridges where nearly all snow has been blown off. HS differences reach up to 5 m within only a few meters in horizontal distance. Artificial terrain features such as hiking paths and the edges of the ski track can easily be identified in the HS map. The gray features on the top and on the left side are the station building of the chairlift Brämajet and its masts. This visualization highlights the role of wind in combination with small terrain features for the spatial variability of HS.

The mean HS distribution classified by the terrain exposions confirms the visual impression that the south facing slopes have much lower HS values than the north facing slopes (Fig. 7). Also the standard deviation of the mean HS shows a tendency to be smaller at southern exposions (SE, S, and SW). This slope aspect analysis was performed on the snow-free DSM, which was resampled to 1 m to filter out small exposion changes. Such statistical evaluation enables a more detailed analysis of mountain HS distribution on local to regional scale.

The comparison of the photogrammetric HS with manual HS measurements results in a RMSE of 0.15 m and a very high correlation coefficient of $R^2 = 0.99$ (Fig. 8). The

**TCD**

doi:10.5194/tc-2015-220

**Mapping snow depth in alpine terrain with unmanned aerial systems (UAS)**

Y. Bühler et al.

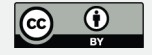

**TCD**

doi:10.5194/tc-2015-220

**Mapping snow depth in alpine terrain with unmanned aerial systems (UAS)**

Y. Bühler et al.

photogrammetric HS values are, on average, 0.11 m lower than the manual measurements. The summer vegetation can at least partly explain difference, as dense grass and small bushes cover the peak of Brämabühl. The comparison of the standard deviations within a reference plot results in a mean deviation of 0.03 m; the RMSE is 0.06 cm.

These results confirm the high accuracy of the photogrammetric HS measurements we found at the Tschuggen test site.

## 5 Discussion

Based on the experience gained at the two presented test sites, the following key points require a more detailed discussion because they are crucial for the application of UASs in high alpine terrain.

### 5.1 UAS applied in high alpine terrain

Steep terrain, high altitudes, low temperatures and often wind speeds of more than $10 \, \mathrm{m \, s^{-1}}$ are typical for high alpine regions. To successfully apply UASs, platform and sensor must be able to handle such conditions and have to be easily transportable in a backpack on foot or on skis. The key limitation of the applied Falcon 8 UAS is the comparably short flight time of 6 to 10 min with one battery at elevations above 2000 m a.s.l. This also limits the range of the UAS. As a consequence, the pilot position has to be close to the area of interest, which is often difficult or even impossible for example if snow avalanche release zones have to be mapped. A big advantage of a multicopter UAS is that they can be started and landed by hand, which is the appropriate starting/landing procedure we apply in alpine terrain. This is in contrast to the application of winged UASs, which require large flat areas to safely land but such areas are typically missing in high alpine regions. Cold temperatures of down to −30 °C are a major problem for battery transportation. As soon as the battery is deployed and running in the UAS there is self-heating. Therefore it is critical that the batteries are transported

in a heated environment for example close to the body, otherwise they will lose a big part of their performance before taking off significantly reducing the already short flight time. On the other hand, our experiences with the UAS regarding high wind speeds were surprising. Even under foehn conditions with gusty wind speeds up to $20\,\mathrm{m\,s^{-1}}$

5  the acquired imagery was of high quality and the flight plan and its specific overlap could be accomplished. Fixed-wing UAS achieve significantly longer flight times per battery (20–60 min), but are less stable in windy conditions, are less easy to transport and to fly and they need gentle terrain to land. In our opinion, this limits their successful application in alpine terrain considerably in particular on missions in alpine terrain.

## 5.2 Photogrammetry on snow covered terrain

For a long time, photogrammetry on snow-covered terrain was considered unfeasible, due to low contrast, a limitation only recently overcome as highlighted in current studies (Bühler et al., 2015a; Lee et al., 2008; Nolan et al., 2015). However, the smoother the snow surface is, the harder it gets for the structure-from-motion software to iden-

tify meaningful matching points. This gets obvious if we look at homogenous areas within the hillshade DSM at shadowed and at well-illuminated snow covered locations (Fig. 9). In shadowed areas (e.g. shadow of the chapel tower) the clearly visible noise introduced into the DEM gets amplitudes of up to 0.40 m. In the bright, very homogenous areas the noise shows amplitudes of up to 0.15 m. This indicates that a fresh

snow surface is less suitable than an older, weathered surface. But due to strong winds and large differences in radiation, alpine snow surfaces develop detectable features such as sastrugis or wind ripples already during or very quickly after fresh snowfall. However, very homogenous snow surfaces occur only within very small parts of our test sites.

Additional problems occur if reflections of the sun on the snow saturate the camera sensor. Therefore it is recommended that the camera exposure time is properly set and the imagery is stored in raw format using the full bit depth of the sensor, typically 10 to 14 bits. Standard JPEG image compression, which is the default storage setting

Discussion Paper | Discussion Paper | Discussion Paper | Discussion Paper |

**TCD**

doi:10.5194/tc-2015-220

**Mapping snow depth in alpine terrain with unmanned aerial systems (UAS)**

Y. Bühler et al.

for most cameras, is limited to 8 bits storing only 256 gray scale values per band. To acquire an optimal contrast on homogenous snow surface we recommend using RAW image storage format with 12 bit. However, further investigations have to quantify the benefit of 12 bit image storage over the 8 bit JPEG compression on snow covered areas.

As snow absorbs more energy in the near infrared NIR part ($\lambda \approx 760$–$2500$ nm) of the electromagnetic spectrum than in the visible part ($\lambda \approx 400$–$700$ nm) and the reflection is sensitive to snow grain size (Warren, 1982) at the snow surface, additional features are expected to be discriminated if NIR data can be used (Bühler et al., 2015b). However, further studies have to investigate the real benefit of NIR bands for photogrammetric HS mapping in more detail. This might only be significant if multi-imager cameras with narrow NIR bands and simultaneous band acquisition are applied.

## 5.3 Orthorectification

Exact relative georeferencing (coregistration) between the two DSMs is essential for correct HS calculation (snow-covered DSM minus snow-free DSM). Even small shifts in $x$ and $y$ can lead to large differences in $z$ direction on steep terrain. The following referencing approaches exist:

a. absolute referencing with artificial GCPs measured with differential GNSS;

b. relative referencing with natural GPSs that are well visible in the snow-free and the snow-covered imagery;

c. absolute referencing of one DSM with differential GNSS and then relative referencing of the second DSM by identifying well visible points in the second DSM.

A major drawback of method (a) is that all reference points have to be manually deployed and measured with differential GNSS devices to achieve accuracy in the range of centimeters to a decimeter. They should be distributed equally over the entire area of interest and all elevation bands. In high alpine terrain this is often not possible for

**TCD**

doi:10.5194/tc-2015-220

**Mapping snow depth in alpine terrain with unmanned aerial systems (UAS)**

Y. Bühler et al.

**TCD**

doi:10.5194/tc-2015-220

**Mapping snow depth in alpine terrain with unmanned aerial systems (UAS)**

Y. Bühler et al.

example due to avalanche danger. The methods (b) and (c) exclude the possibility of a potential GNSS shift but are only applicable if areas with distinct terrain features exist that are not covered by snow. This was the case at our test sites but might not be feasible in winters with exceptionally high amounts of snow. The referencing strategy has to be evaluated carefully prior to a UAS HS mapping campaign. A direct matching of the snow-covered to the snow-free point cloud (Gruen and Akca, 2005) is not feasible as the terrain shows large differences over most parts due to the snow cover.

## 5.4 Underlying vegetation

Within the accuracy range of 0.05–0.15 m, the vegetation at the base of the snow cover has a significant influence on the results. At the test site Tschuggen small bushes, mainly alpine rose, juniper and erica, rise up to 0.50 m above ground in summer (Fig. 10a). In winter they are pressed down to the ground by the snowpack but form a snow-free layer at the bottom of the snowpack which can have a depth of a few centimeters to decimeters (Feistl et al., 2014). This leads to a systematic underestimation of HS mapped with photogrammetry (snow-free DSM is too high) as well as a systematic overestimation of HS measured manually with the avalanche probe because the probe penetrates the snow-free bottom layer and sometimes even the first layers of the ground. The "real" HS is most probably a value between the manual probe and the photogrammetric measurements. High grass on the other hand is usually pressed down to the ground completely only leaving a snow-free layer of less than some centimeters (Fig. 10b). This makes the probe measurements more reliable but can falsify the photogrammetric measurements significantly if the grass is high during the snow-free data acquisition. Alpine meadows should therefore be surveyed right after mowing or late in autumn while the grass is low. From our experience it is very difficult to correct the photogrammetric HS based on underlying vegetation because the elevation differences vary very much within short distances. A possibility might be to apply a vegetation classification based on the orthophotos to correct the underestimation of HS in areas with many bushes. But there is a high risk to introduce new errors and this possibility has to

**TCD**

doi:10.5194/tc-2015-220

be investigated in more detail in the future. Photogrammetric HS mapping is impossible above and around trees as trees are nearly always moved by wind and the resulting ambiguous tree top positions interfere with image matching. Additionally areas below trees are not visible in the nadir imagery. Therefore laser scanning, measuring first and last returns or even full wave form signals, is still the best choice for investigations where trees play a major role (Moeser et al., 2015).

## 6 Conclusions

UAS-based digital photogrammetry is able to map the spatial variability of alpine HS with accuracies of 0.07 to 0.15 m RMSE compared to traditional manual measurements with avalanche probes. These accuracies are in the same range as HS measurements acquired by terrestrial laser scanning (Deems et al. 2013) and reported in the manned airplane based study by Nolan et al. (2015) and the UAS based study by Jagt et al. (2015). It is significantly better than the RMSE of 0.30 m reported by Bühler et al. (2015a), using an ADS80 survey camera mounted on a manned airplane, but can only cover considerably smaller areas. Fixed-wing UAS, flying at high altitudes above ground, would be able to cover larger areas of several square kilometers. Future investigations have to clarify how accurate the results from such platforms can get as the spatial resolution of the input imagery is worse and the results might get much more affected by wind.

UASs enable fast, flexible, repeatable and detailed analysis of the spatial distribution of the mountain snow cover. We successfully applied a complete photogrammetric workflow at a sheltered test site at the valley bottom (Tschuggen) and at an exposed test site at a mountain top (Brämabühl) mapping extreme HS variability of up to 5 m within less than 3 m distance, confirming the important role of wind and terrain features on HS distribution in alpine regions (Mott et al., 2010).

A key to robust photogrammetric HS measurements is the accurate co-registration of the snow-free and the snow-covered digital surface models (DSM). Even small shifts

**Mapping snow depth in alpine terrain with unmanned aerial systems (UAS)**

Y. Bühler et al.

Title Page

Abstract · Introduction

Conclusions · References

Tables · Figures

|◄ · ►|

◄ · ►

Back · Close

in $x$ and/or $y$ direction can lead to large shifts in z in particular within steep terrain. To avoid shifts introduced by global navigation satellite system measurements (GNSS) we propose to reference the snow-covered DSM directly on the snow-free DSM. But this is only possible if snow-free areas exist, that contain well visible point- or linear features. Another important point is that alpine vegetation, such as bushes and tall grass, lead to a significant overestimation of snow-free DSM elevations, resulting in underestimated HS values. This can introduce errors in HS values of up to 0.50 m.

We identify numerous promising applications where UAS have not yet been applied:

– precise water resource prediction for hydropower and flood warning in alpine catchments (Jonas et al., 2009);

– validation of snowpack and snow hydrology models (Bartelt and Lehning, 2002; Mote et al., 2003);

– survey of snow distribution in ski resorts to improve the track management (Damm et al., 2014);

– precise documentation of specific avalanche release and deposition to validate and calibrate numerical avalanche simulations (Christen et al., 2010) and to generate precise, up-to-date DSMs e.g. after an avalanche event blocks a channel, as base for such simulations (Bühler et al., 2011);

– identification of representative locations for automated weather stations (Grünewald and Lehning, 2015);

– survey of avalanche defense structures to prevent ineffectiveness due to potential overfill (Margreth and Romang, 2010);

– ideal positioning of artificial avalanche release trigger points (Stoffel and Margreth, 2009);

– identification of wind blown snow packets prone to snow avalanche release (Schweizer et al., 2008).

**TCD**

doi:10.5194/tc-2015-220

**Mapping snow depth in alpine terrain with unmanned aerial systems (UAS)**

Y. Bühler et al.

In our opinion the number of investigations applying UAS based digital photogrammetry will rise quickly within the next years and change the way of collecting spatial information within alpine terrain sustainably.

*Acknowledgements.* Parts of this work was supported by the Austrian Academy of Sciences (ÖAW) under the project RPAS4SNOW.

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

**Table 1.** Technical specifications of the Falcon 8 UAS.

| UAS type | V-form octocopter |
|---|---|
| Dimensions | 770 × 820 × 125 mm |
| Engines | 8 electrical, brushless (sensor less) motors |
| Rotor diameter | 8″ (∼ 0.20 m) |
| Number of rotors | 8 |
| Rotor weight | 6 g |
| Empty weight | 1.1 kg |
| Max. take off weight | 2.3 kg |
| Max. payload weight | 0.8 kg |
| Max. flight time per Battery | 12–22 min |
| Max. range | 1 km |
| Tolerable wind speed | 12–15 m s$^{-1}$ |
| Navigation sensors | AscTec Trinity (IMU, barometer and compass) |
| | AscTec High-Performance GPS (GNSS) |
| Max. airspeed | Manual mode 15 m s$^{-1}$ |
| | Height mode 15 m s$^{-1}$ |
| | GPS mode 4.5–10 m s$^{-1}$ |
| Max. climb/sink rate: | Manual mode 6–10 m s$^{-1}$ |
| | Height mode 3 m s$^{-1}$ |
| | GPS mode 3 m s$^{-1}$ |
| Wireless communication | 2 independent (diversity) control/data links 2.4 GHz FHSS link (10 to 63 mW) |
| | 1 analogue diversity video receiver 5.8 GHz (25 or 100 mW) |
| LiPo battery | PP 6250, 3 Cells 6250 mAh (∼ 426 g) |

Discussion Paper | Discussion Paper | Discussion Paper | Discussion Paper

**TCD**

doi:10.5194/tc-2015-220

**Mapping snow depth in alpine terrain with unmanned aerial systems (UAS)**

Y. Bühler et al.

**Table 2.** Data acquisition parameters for Tschuggen.

| Acquisition date | Images | Covered area | Mean flight height above ground | Average points per m$^2$ | Reference measurements |
|---|---|---|---|---|---|
| 11 March 2015 close to peak of winter | 43 | 57 000 m$^2$ | 97 m | 772 | 12 plots (60 single points) |
| 24 April 2015 snow melt ongoing | 55 | 87 000 m$^2$ | 126 m | 469 | 19 plots (95 single points) |
| 12 March 2015 snowmelt nearly completed | 55 | 91 000 m$^2$ | 130 m | 439 | 19 plots (95 single points) |
| 28 September 2015 completely snow free | 99 | 128 000 m$^2$ | 113 m | 563 | – |

Discussion Paper | Discussion Paper | Discussion Paper | Discussion Paper

**TCD**

doi:10.5194/tc-2015-220

**Mapping snow depth in alpine terrain with unmanned aerial systems (UAS)**

Y. Bühler et al.

**Table 3.** Data acquisition parameters for the Brämabühl test site.

| Acquisition date | No. of images | Covered area | Mean flight height above ground level | Average points per $m^2$ | No. of reference measurements |
|---|---|---|---|---|---|
| 14 April 2015 close to peak of winter HS accumulation | 85 | 285 000 $m^2$ | 157 m | 274 | 22 plots (110 single points) |
| 21 September 2015 completely snow free | 274 | 363 000 $m^2$ | 133 m | 386 | – |



**TCD**

doi:10.5194/tc-2015-220

**Mapping snow depth in alpine terrain with unmanned aerial systems (UAS)**

Y. Bühler et al.

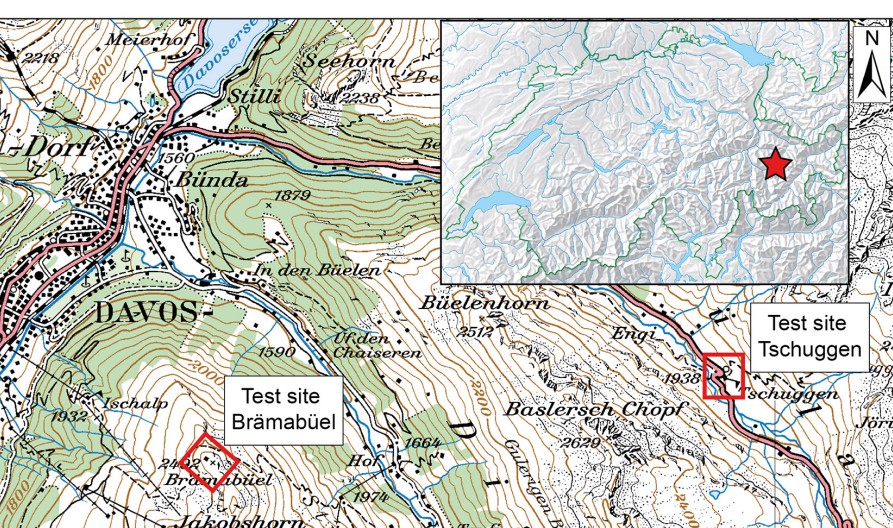

**Figure 1.** Location of test sites Tschuggen and Brämabüel close to Davos, Switzerland, Pixmap© 2015 swisstopo (5 704 000 000), reproduced by permission of swisstopo (JA100118).



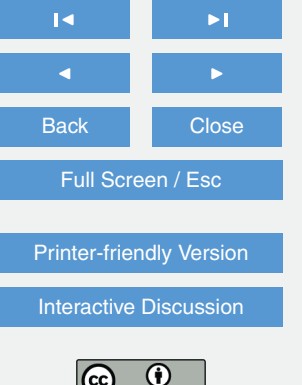

**Figure 2.** Orthophotos of the four different data acquisitions at Tschuggen depicting the change in snow coverage overlaid by the locations of the manual HS measurements and the applied ground control points.

**TCD**

doi:10.5194/tc-2015-220

**Mapping snow depth in alpine terrain with unmanned aerial systems (UAS)**

Y. Bühler et al.

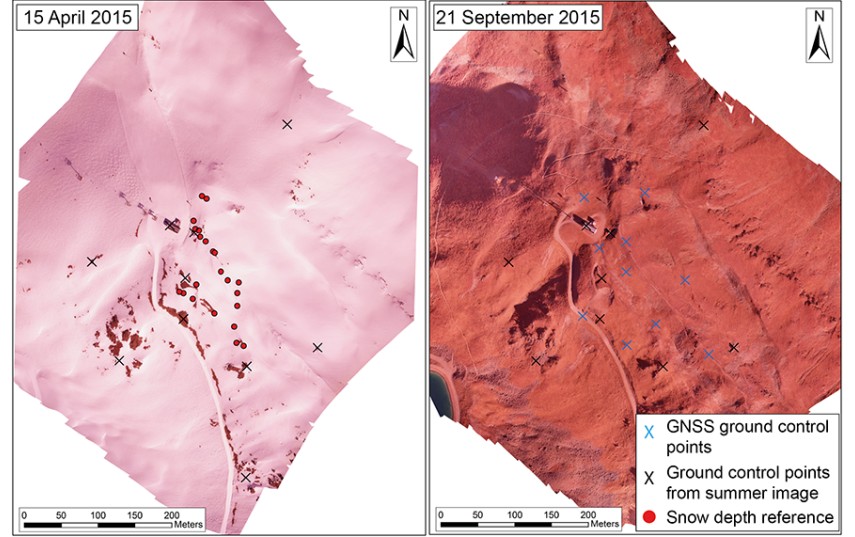

**Figure 3.** Near infrared orthophotos snow-covered (left panel) and snow-free (right panel), acquired over the Brämabühl test site including the applied ground control points and reference HS measurements.

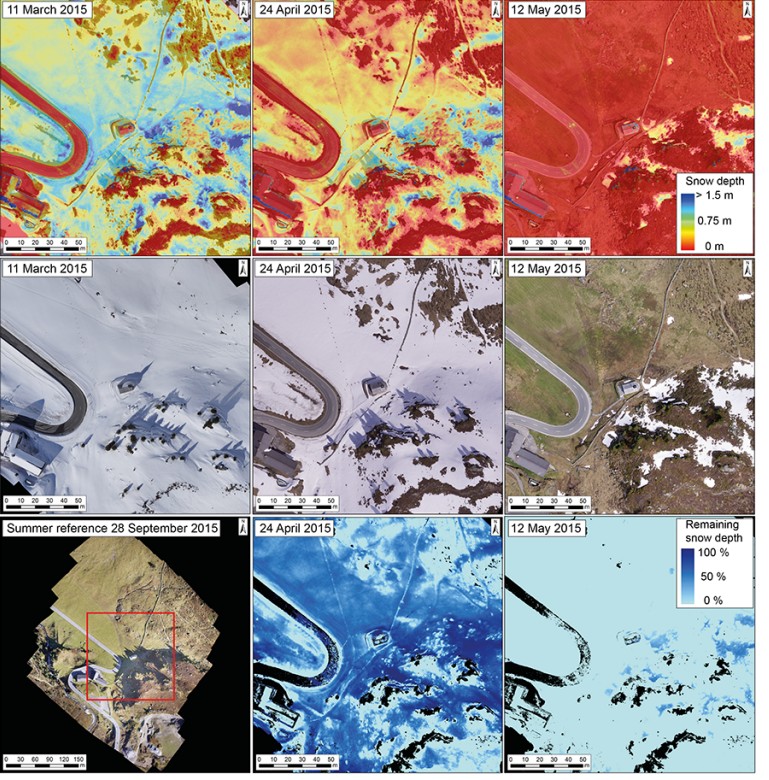
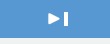
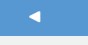
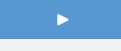
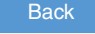


**Figure 4.** HS maps (top panels) and corresponding orthophotos (middle panels) of the area around the chapel in the center of the test site. At the bottom the orthophoto of the snow-free reference (bottom left panel) and the spatial distribution of melt rates as percentage of remaining snow compared to the peak of winter (11 March 2015) are depicted. Black areas are no data values.

Discussion Paper | Discussion Paper | Discussion Paper | Discussion Paper |

**TCD**

doi:10.5194/tc-2015-220

**Mapping snow depth in alpine terrain with unmanned aerial systems (UAS)**

Y. Bühler et al.

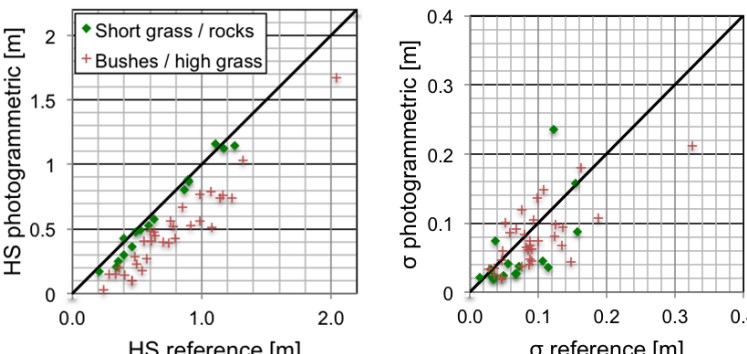

**Figure 5.** Statistical evaluation of the HS measurements (left panel) and the standard deviations $\sigma$ of HS within a specific reference plot (right panel). The overall correlation coefficients $R^2$ is 0.84 ($R^2 = 0.98$ for the class *short grass/rocks* and $R^2 = 0.92$ for the class *bushes/high grass*).

Discussion Paper | Discussion Paper | Discussion Paper | Discussion Paper | Discussion Paper |

TCD

doi:10.5194/tc-2015-220

**Mapping snow depth in alpine terrain with unmanned aerial systems (UAS)**

Y. Bühler et al.

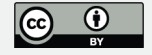

Discussion Paper | Discussion Paper | Discussion Paper | Discussion Paper

**TCD**

doi:10.5194/tc-2015-220

**Mapping snow depth in alpine terrain with unmanned aerial systems (UAS)**

Y. Bühler et al.

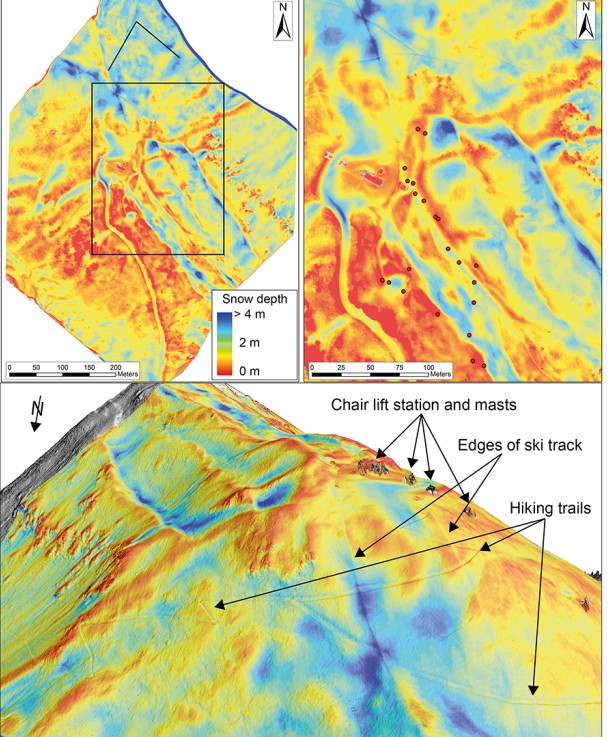

**Figure 6.** Overall HS map of the Brämabühl test site (top left panel) and close-up of the central part (top right panel). The locations of the reference plots are displayed as red circles. 3-D view of the HS draped over the hillshade of the snow-free DSM looking from north to south (bottom panel).

Discussion Paper | Discussion Paper | Discussion Paper | Discussion Paper | Discussion Paper |

**TCD**

doi:10.5194/tc-2015-220

**Mapping snow depth in alpine terrain with unmanned aerial systems (UAS)**

Y. Bühler et al.

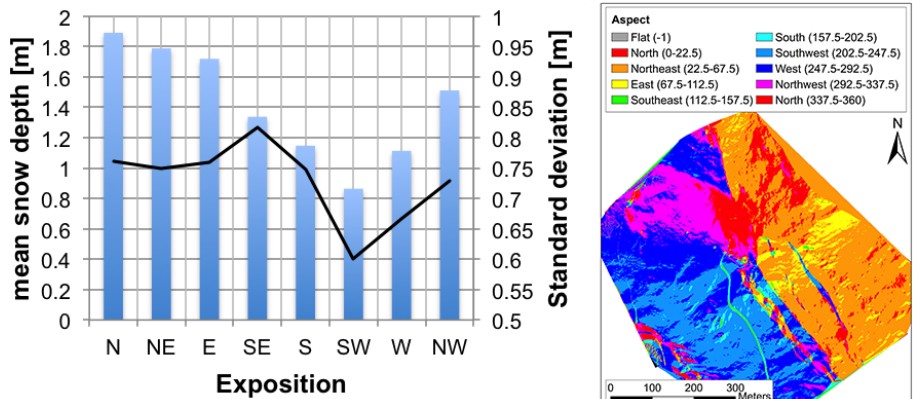

**Figure 7.** Statistical evaluation of the mean HS and its standard deviation, classified by the exposition (left panel) and exposition map (right panel).

TCD

doi:10.5194/tc-2015-220

**Mapping snow depth in alpine terrain with unmanned aerial systems (UAS)**

Y. Bühler et al.

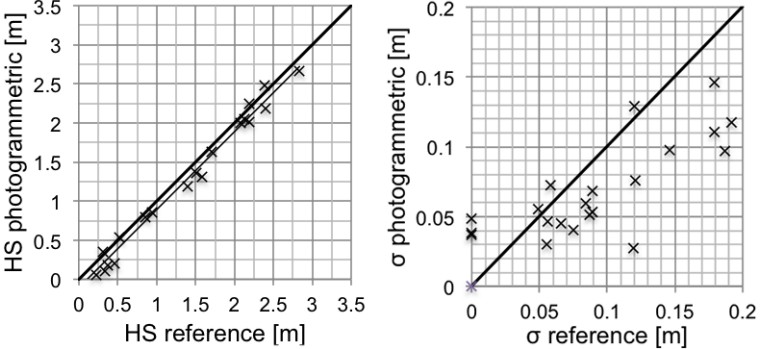

**Figure 8.** Statistical evaluation of the HS measurements (left panel) and the standard deviations $\sigma$ of HS within a reference plots (right panel).

Discussion Paper | Discussion Paper | Discussion Paper | Discussion Paper

**TCD**

doi:10.5194/tc-2015-220

**Mapping snow depth in alpine terrain with unmanned aerial systems (UAS)**

Y. Bühler et al.



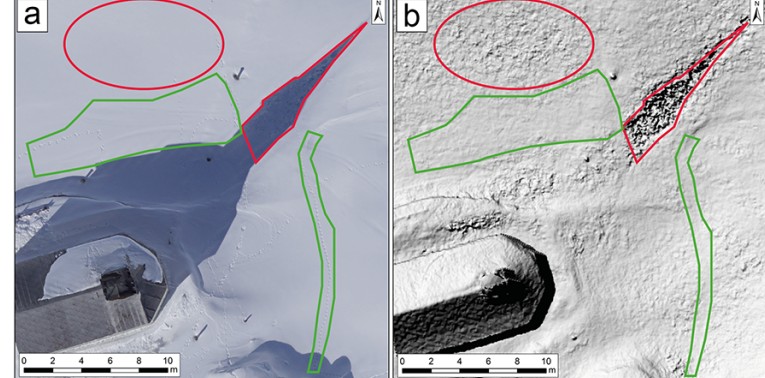

**Figure 9.** Winter orthophoto of the area close to the chapel within the test site Tschuggen **(a)** and hillshade of the derived DSM **(b)**. Areas in red show very homogeneous snow surfaces either in cast shadow or nearly saturated areas. Areas marked in green are areas with better contrast at the snow surface due to tracks of animals or wind features.

Discussion Paper | Discussion Paper | Discussion Paper | Discussion Paper |

**TCD**

doi:10.5194/tc-2015-220

**Mapping snow depth in alpine terrain with unmanned aerial systems (UAS)**

Y. Bühler et al.

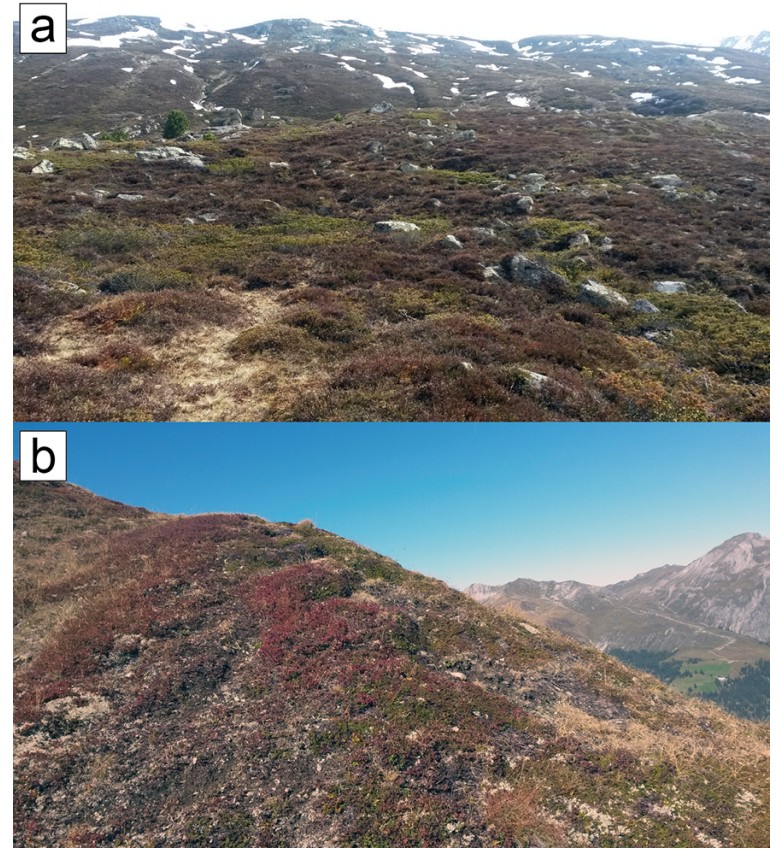

Figure 10. **(a)** Photograph of the bushes that rise up to 0.50 m above ground and patches of low grass at the test site Tschuggen. **(b)** Photograph of the shallow vegetation at the test site Brämabühl with maximum elevation of approximately 0.15 m.