# Peer review of "Mapping snow depth in alpine terrain with unmanned aerial systems (UAS): potential and limitations"

_The Cryosphere, 2015_

## Referee Comment (RC1) · M. Nolan (Referee) · 9 Mar 2016

"UASs enable fast, flexible, repeatable and detailed analysis of the spatial distribution of mountain snow cover". This sentence from the paper describes the essential findings of the work, though I would add "over several hectare areas" to improve the accuracy of that description. Towards these ends, the authors have conducted sound scientific experiments that are well supported and described, and I believe their work deserves to be published. The only scientific analysis I found lacking was an analysis of the repeatability of their system – measuring the same location twice on the same day (or a snow-free road on two different days) and seeing how close the measurements are to each other; that is, determining the noise level of their system, and it seems they have data in hand to do this.

[Figure]

The authors are clearly strong supporters of UAS technology, and I applaud and encourage their efforts to push the boundaries of this technology towards such an important scientific subject. However, the paper reaches well beyond the scope of its scientific findings to make claims about the implications or justifications of this work without support for those claims. I found two categories of such claims. First are claims that UAS are somehow more cost effective to use than manned aircraft. Though I readily admit my bias, as a scientist on a budget I would not be using a manned aircraft to measure snow pack photogrammetrically if I believed this to be true. These claims need to either be removed or validated through an actual economic analysis, and this analysis needs to at least encompass variables such as region of the world, full costs for manpower, and areal coverage. For example, I can map 100 km2 at 10 cm GSD in an hour in my manned aircraft and I can do so over steep, dangerous terrain without risk being caught in an avalanche, for a total of perhaps 4-5 man-hours of field effort. By comparison, the UAS work in this paper failed to demonstrated that it could map more than 1 km2 in a day's work for several people – though its direct costs may be much less, how much salary time would it take a 2-3 man team to map 100 km2? Perhaps there are economics that I don't understand and I am happy to be educated, but in any case these statements require justification before manned aircraft can be summarily dismissed in favor of UAS due to cost. This leads to the second category of unsupported claims regarding future use of UAS for the purpose of wide-area mapping. The conclusions, for example, list 8 future uses of UASs, only one of which the authors have shown any support for within the paper. For example, claims that a UAS can make "precise water resource predictions for hydropower and flood warning in alpine catchments" – that is, that they can map 100s of km2 – have no support in the paper, and indeed the paper admits several times that the limited flight times of 10-20 minutes are a major hindrance to their research in even small areas. As another example, staying in line of sight of the UAS means that the pilots must travel essentially through the dangerous avalanche terrain they claim their UAS can measure. If the authors want to assert these uses, then more validation and description is required that their system is

capable of it. I'm enthusiastic about the potential uses for this technology, but I don't see that the actual uses are highlighted here.

Thus overall I think the paper would be substantially improved by changing the wrapper placed around their work and rewriting it to focus on the useful results they found and their true significance – they have shown that they can measure several hectare areas in a variety of terrain types at very high spatial resolution and very good accuracy and this will benefit many types of studies that are currently hampered by the lack of such measurements. There are plenty of such applications, no need for touting these as a replacement for manned aircraft in those many applications where manned aircraft are much more cost effective (like large area mapping) and much safer. The text could use a bit of cleanup but is overall well written and the science seems well done, supported, and verifiable, though as stated earlier a repeatability spec would improve it further.

Specific Comments Abstract Line 1: Not really a topic sentence. Best to get as much of the who, what, where, why, and when out in the first sentence, but this is personal preference. Line 2: No need for "(HS)" as you don't use it again within the Abstract Line 3: "Nowadays" is an odd word here Line 6: This sentence is not quite accurate or meaningful, as 'dense' is not defined well enough to evaluate it. A dense enough network could be devised for any locale, the question is really whether it is feasible to implement. Line 10. The implication by saying 'costly' is that UAS are cheaper. Remove, or support in the paper. Line 15. Again, either provide an analysis in the paper that UAS are "comparatively cost effective" or remove the statement. Similarly about the next part of the sentence for use in "otherwise inaccessible terrain" as this was not supported in the paper as all the sites used were easily accessible, and the paper actually recognizes this as a limitation. Line 21. RMSE of "snow depth values"? Do you mean residuals between the measurement types? Or a mean snow depth? Or? Line 24. Again, remove cost effective or justify, and clean up the end of the sentence a bit.

Introduction I believe in this section some clear mention should be made of the true

roles that UAS can play today in terms of areal coverage and contrast this with what manned aircraft can do. I use both, but I only use a UAV when I'm already on the ground somewhere. This is the place for an economic justification for the use of UAVs over manned aircraft, if there is one. Flying a manned aircraft to a remote location to drop off a team to use a UAV in a tiny area makes little scientific sense for most applications and costs more. But if you have a road or trail system through a mountain range with huts that serve spaghetti every 5 miles and you have no budget at all then using a UAV to map small areas nearby may make some sense economically. Or however you think about it, just be explicit about your claims. Please also see Nolan and Deslauriers 2015 currently in Cryosphere Discussions, where we map snow depth over the tallest and most remote peaks in the US Arctic using a manned aircraft. Here we show that we can truly map avalanche danger, cornice development, gully filling, etc, not as some future possibility but as true examples of our current capabilities. While we did not discuss economics much there, the ability to map snow depth on a big chunk of a mountain range located 350 miles away in a single flight is something that UAS will never be able to do at any cost, and this is worth bearing in mind in this paper, especially since UAS are banned in most US federal lands. Here also some mention should be made of what sorts of projects that a UAS can actually do better than can be done from a manned aircraft; if there are none, this should be stated (I think there are).

Line 27. Qualify this claim further. Do you mean the equipment is very expensive? Or commercial acquisitions? If a University or lab already owned own, ts not very expensive to operate. Page 4, Line 2. Again, provide support for "cost-effective"

Methods Page 5, Line 20. Near infrared is mentioned several times throughout the paper as having advantages on snow, but I found no results of this UAS work that supported this. Perhaps I missed it, so this should either be emphasized further or this discussion toned down.

Page 7, Line 13. The quality setting is directly related to resolution used in the calculations: ultra high uses each pixel individually, High uses 2x2 pixels, Medium 3x3, etc. The filtering is mostly necessitated by parallax caused by motion and match point errors I believe. This doesn't need to be mentioned in the paper, just commenting. Page 7, Line 25. This sentence is confusing. It says two "well accessible" sites that are "typical locations" – does this mean most sites in these mountains are easily accessible? This relates directly back to claims earlier of being able to work in inaccessible locations. Page 8, Line 14. Do you have support for this claim of being a good compromise? I think its true, but it should be supported when stated like this. Page 8, Line 16. I don't see anywhere in the paper or tables specs on the GPS accuracy of the UAV position? It strikes me that the 'older' version may actually be better than the newer one, because if the UAV stabilization on a location, its positional accuracy may be improved simply because there the timing error is reducing (if the position uses the camera's exif data in integer seconds). Have you explored whether the old and new methods give the same results? Page 8, Line 21. The word 'selected' is repeated. Page 8, Line 25. How was orthoimage accuracy measured? By eye in comparison to photo-identifiable GCPs? What does the Z value mean in terms of an orthophoto? Page 9, Line 17. I'm confused about the use of the NIR imagery. From figure 3, it looks to me that the NIR shows less detail than the other. The text says NIR is 'expected' to be better – well, was it? Page 9, Line 26. I'm confused about the use and necessity of GCPs in this study. Are these being used in the bundle adjustment at all, or just for validating the results? A clear statement needs to be made about this. Page 11, Line 7. I'm confused as to what this classification is doing? Also, why set negative snow depths to zero? There is clearly snow there, so its not zero.

Page 13, Line 17. Here range and pilot positioning are discussed as being limitations. There's nothing wrong with this, it is what it is. UAVs have a place, but that place is not wide area mapping as can be done from manned aircraft. As stated earlier, I think these limitations need to be discussed in the abstract and introduction, so as not to give the reader false expectations about what UAVs are capable of, but I also don't feel that limitations are something to be ashamed of, different tools for different jobs.

Page 15, Line 6. Again, its not clear whether the research of this paper demonstrated anything regarding NIR superiority, so its not clear to me what this paragraph's purpose is. Page 15, Line 19. "GPSs" is used when I think "GCPs" are meant. I found these 3 options confusing and I'm not sure what any of them mean. What's the difference between "a" and "c"? Why are GCPs needed at all – why not just co-register them and ignore the realworld coordinates? There is also a better option – just use the manual probe depths for co-registrations. This is a great advantage for UAVs – you are going to be standing in your field area anyway, so you have opportunity to probe, and then just match the UAV snow depths to those manual probe measurements, at locations where vegetation is minimal. Further, this sort of registration is only required in the first place because the on-board GPS is not accurate enough to directly georeferenced the data accurately enough for this application; this should be discussed and mentioned for future development. That is, if your photo positions had < 1cm accuracy, your maps would too, and this is a possibility for slow moving UAVs even today.

Page 16, Line 9. Accuracy of what? Page 17, Line1. This is far overstated. UAVs have particular trouble with tree motion because they are such high resolution GSD (or TSD in this case...). From a manned aircraft, tree motion has a negligible effect on results in the 30-50 cm GSD range, and I have tons of data at 5-10 cm GSD within forests. Whether the area beneath the tree is visible depends on the tree – our black spruce are quite skinny, and our birch tree lose their leaves in winter allowing us to map the ground beneath clearly.

Conclusions This section needs a rewrite, as there is a lot of Discussion mixed in with actual review of results and findings. For example, the fixed wing UAS discussion. See also my earlier comments regarding using the manual probe depths for co-registration and to eliminate the last paragraph/list describing uses that UAVs are not capable of currently (and probably never will be) or justify these claims more fully, but in any case move to Discussion. I believe a list of true potential uses for the system that the authors used to measure snow depth (that is, snow depth with tiny GSD over small areas) would

be a great idea, but this should placed in the Discussion.
* * *

---

## Author Comment (AC1) · 16 Mar 2016

Dear Matt Nolan,

We thank you for your positive feedback and the constructive comments.

We do not at all intend to talk down the value of manned aircraft for snow depth mapping over large areas. We are ourselves strong supporters of this method (e.g. Bühler et al. 2015). However, to cover small areas, I am positive that UAS are more economic than airplanes in most countries around the world. The situation you have in Alaska is very special. I was really jealous to see all the nice airplanes in the gardens outside Anchorage. This is really a dream come true for every remote sensing guy. However, in Switzerland and most European countries, it is quite time-consuming and costly to get an airplane and acquire the necessary flight permissions. This is

additionally hampered by very restrictive ATC-regulations for small airplanes in Central European airspace and the use of alpine airports. Repetition flights are nearly impossible to do, we collected some quotations on that. On the other hand, the regulations to fly UAS are different for every country and sometimes even for different states and they are changing quickly. Therefore it is impossible to list all the different regulations. We confine us to give a brief description of the most important regulations for Switzerland in the paper. However, as you suggest, we amended the proposed addition"over several hectare areas"to the sentence in the conclusions to make this clearer. And you are right, compared to North America, the Swiss Alps have a very dense infrastructure (roads, railways and huts that serve spaghetti and fondue), which makes it much easier to deploy UAS, because you can reach most spots of interest quiet easily. On the other hand it is very difficult and expensive to get an airplane with flight permission on short notice.

We will include an assessment of the repeatability by analyzing the snow free road at the Tschuggen test site at all 4 flight dates. Thank you for this valuable suggestion. We add the following paragraph to chapter 4.1: "To assess the repeatability of the UAS HS mapping, we analyze the altitude deviation of the different DSM for 10550 grid cells on the snow-free road. The calculated RMSE values compared to the summer DSM (28 September) are 0.093 m (11 March), 0.052 m (24 April) and 0.045 m (12 May). This indicates that the noise of the method is smaller than 0.1 m."

Specific comments:

P1L1: The first sentence intends to give a broad picture why snow depth information is important. We therefore want to keep this sentence.

P1L2: We use the abbreviation HS for snow depth consequently through the paper, therefore we would like to already include it in the abstract.

P1L3: We change Nowadays into Currently

P1L6: We change the sentence to "even a dense measurement network like in Switzerland with more than one measurement station per 10 km2 in average "

P1L10: We add "in most countries" to consider the situation in Alaska.

P1L15: We change otherwise inaccessible not accessible from the ground. Not all parts of the Brämabühl test site were accessible. The steep north-facing slope is very prone to avalanche danger and cannot be accessed during most days in winter.

P1L21: We added "compared to manual snow-depth measurements" to clarify this

P1L24: We change the sentence as following "This new measurement technology opens the door for efficient, flexible, repeatable and cost-effective snow depth monitoring over areas of several hectares for various applications."

Introduction: We put the proposed discussion on UAS capabilities to the discussion section. Are you sure you are able to map avalanche danger as you write? Your impressive results in Nolan and Deslauriers (2015) show the mapping of cornices and the filling of terrain features. However I do not see that you map avalanche danger.

P2L27: We listed three different quotations of airborne LiDAR- and Photogrammetric data acquisition in Bühler et al. (2015). It is clear, if you already have a full system including airplane and pilot, that the costs are considerably reduced. However, we doubt that there are lots of users who have all this equipment available (especially in Europe).

P4L2: please see answer above

P5L20: We cite two papers demonstrating the advantage of near infrared bands. However, in this study we are not able to investigate this topic in detail. Therefore we just mention the potential by citing. We are right now working on a detailed study to quantify the benefit of near infrared bands.

P7L13: We think this is interesting information for readers who want to apply SfM and

we would like to keep it in the paper.

P7L25: We usually choose locations for our investigations that are well accessible. We would have a lot of other areas, which are not as well accessible. But the terrain characteristics of the choosen test sites are representative for a lot of areas. We change typical locations into " that represent typical terrain characteristics in high-alpine environment " to clarify.

P8L14: We did several tests with different overlaps. Our experience showed that 70% is a good compromise. We add "from our experience with different overlaps we conclude that"to clarify why we get to this conclusion.

P8L16: We do not use the position of the cameras recorded by the UAS GNSS as the accuracy is estimated to approximately 3 m, which is insufficient for snow depth mapping application. We only use the UAS GPS to fly the lines defined in the flight planning software. Unfortunately the producer of our UAS gives no details on the applied GNSS sensor.

P8L21: we replace selected with applied

P8L25: This is the orientation accuracy, not the accuracy of the orthoimagery. These values are calculated from the applied GCPs. We change orthorectification to "orientation".

P9L17: See answer to P5L20. For this study we do not have simultaneously acquired NIR and RGB data but we work on a secondary study where we will have such data available.

P9L26: We mixed up the correct terms. What we use are reference points not control points. We use these points to absolutely orientate the photogrammetric products. We change GCPs to reference points RPs throughout the paper.

P11L7: We classify the are to snow and no snow. Areas not covered by snow are set to HS = 0. Negative values are not changed but due to the color bar limit, they are

depicted red (HS <= 0). We change the sentence to "…. snow-covered areas have been separated from snow free patches using a simple unsupervised classification."to clarify this point.

P13L17: In our opinion the discussion is the right place to discuss this question, as we do. However, we added your suggested changes in the abstract restricting the application of UAS to several hectares areas. As you suggest we move the potential applications from the conclusion to this discussion chapter.

P15L6: In this section we discuss what could potentially improve the results. As stated before we are working on a secondary study quantifying the assumed benefits. We clearly state " However, further studies have to investigate the real benefit of NIR bands for photogrammetric HS mapping in more detail. "

P15L19: Yes this is a mistake. We change GCPs to RPs anyway (see answer P9L26). C is a combination of a and b leading to an absolutely referenced HS map with no offsets in the HS values. As we do not use the UAS GNSS measurements (they are not accurate enough), we must apply RPs, either relative or absolute. We do not understand your proposed approach of matching the HS maps to the probe measurements as we would have a large number different cells with the HS value of the probe (plus minus lets say 5 cm of error). Furthermore we would not be able to assess the achieved accuracy of the HS product if we use the probe measurements for referencing. We add the following sentences to discuss the possibility of very accurate onboard GNSS referencing: "RPs would not be necessary if a very accurate (better than 0.05 m) GNSS system would be available directly on the UAS. First UAS products with such high-accuracy GNSS sensor are already available on the market. However a first investigation by Harder et al. (2016) indicates that the achieved orientation accuracy is not sufficient for snow depth mapping without ground reference measurements."

P16L9: We add "of the HS maps"

P17L1: Here our experience is different. We are not able to reliably map the ground

around and below trees and bushes. It is possible that this effect is less distinct within data with lower GSD. Therefore we want to keep this statement. However we change impossible into "difficult"

Conclusion:

As suggested we move the list of potential applications to the discussion chapter. In our opinion it is not necessary to map entire catchments for water resource prediction or flood warning, but it may be sufficient to map representative subsections which can be done wit UAS. Therefore we want to keep this application in the list even though this can also be done with manned airplanes, as we showed in (Bühler et al. 2015). However we add "small" to alpine catchments.

---

## Referee Comment (RC2) · A. Prokop (Referee) · 29 Mar 2016

First I want to point out that the authors have conducted sound scientific experiments that are well supported and described, and I believe their work deserves to be published. UAS snow depth measurements will be an useful alternative to measure spatial snow depth distributions in the future and I encourage the authors efforts to push the boundaries of this technology towards such an important scientific subject as well. I also understand and would have had probably the same enthusiasm because of the great data presented, however, when M. Nolan summarizes that the sentence: "UASs enable fast, flexible, repeatable and detailed analysis of the spatial distribution of mountain snow cover" describes the essential findings of the work, I have a slight different opinion. UASs enable neither fast nor flexible analysis of the spatial distribution of mountain snow cover as it depends on with what method you compare it to. If you

compare it to manual snow probing you are certainly right, but not in comparison to recent technologies such as laser scanning. UASs need a lot of pre-organizing work, if it comes to snow depth measurements many ground control DGPS measurements (to achieve such an accuracy), and significant post processing time. In comparison to laser scanning not that fast. With being flexible I have the biggest issues, in my opinion UASs are everything but flexible. For commercial purposes (and that counts also for scientific applications) the legal limitations in many countries are significant. Usually you need a proper license to fly above populated areas such as ski resorts and you need to get permission for every flight from the air space administration. For example it took us 1 month of paper work to have an UAS flying on Svalbard (same size and weight). And then you are only allowed to fly in line of sight, which means in practice an area about 500 per 500 m wide, which is exactly what you present in your data. Thinking further on about suitable flying weather and illumination conditions you have in harsh mountain climate conditions such as in maritime coastal mountain weather or arctic weather rather more down days than flying days (strong wind and/or snowfall, very cold and high altitudes (low battery), night, very flat light, etc.)! You present data from 6 measurement days with perfect weather conditions (I assume), therefore it would be interesting what wind, air-temperature as well as air-pressure and cloudiness occurred while completing the measurements. I have seen many researchers UAS crashing due to unfavorable conditions in mountainous terrain! So you have to learn flying an UAS first, before starting to make useful measurements. Furthermore you need significant computing time and rather powerful computer equipment to post process the data and create the DEMs. Thinking about the costs, I have the same opinion as Matt, I do not think it is cheaper to ask a company to deliver a DEM of a snow surface 500 x 500m, using a UAS or a laser scanner. In fact I know 2 companies that charge the same, they just use the laser scanner or the UAS depending on the area they have to measure. If the incident angle is sufficient enough and no shaded areas exist they always use the laser scanner. So I would reduce your statement to what you explain in a later step to something like this: "In particular within flat areas, where

terrain sections behind convex landforms such as hills or moraines cannot be covered, UAS based digital photogrammetry is a promising option for HS mapping in alpine terrain." That hits the application in reality much better in my opinion. That leads me to the applications you have in mind: "precise water resource prediction for hydropower and flood warning in alpine catchments (Jonas et al., 2009)Âż I have the same opinion as Matt, not enough coverage. Same counts for: "validation of snowpack and snow hydrology models (Bartelt and Lehning, 2002; Mote et al., 2003)" Snow pack models yes, but snow hydrology? I think you did a good job to describe it for small catchments. "Survey of snow distribution in ski resorts to improve the track management (Damm et al., 2014)Âż I do not see that at all, track management in ski resorts is made by GPS measurements in real time from snow groomers, which is quiet sufficient, the worker knows immediately how much snow is underneath him (his snow groomer), so I do not see ski resort employees additionally flying around with a UAS (above people?) needing to post process the data, etc. All other applications have also been satisfactorily completed by alternative methods in the same or better accuracy, which are more flexible than UAS, so why using now an UAS (laser scanners scan up to 5000 m nowadays in very fast operation speed, the measurement is basically done in 30 min. all in all) . For sure you can use an UAS for those applications, but there is no improvement to existing methods except for the already mentioned one. So to summarize my maybe too long statement and I just cite Matt: the paper reaches well beyond the scope of its scientific findings to make claims about the implications or justifications of this work without support for those claims. These claims need to either be removed or validated. Thus overall I think the paper would be substantially improved by changing the wrapper placed around their work and rewriting it to focus on the useful results they found and their true significance – they have shown that they can measure several hectare areas in a variety of terrain types at very high spatial resolution and very good accuracy and this will benefit many types of studies that are currently hampered by the lack of such measurements. For the methodology I have a further comment, I really do not understand why you did not validate your data against laser scanning data and

used manual probing instead, see Prokop et al. 2008 (Prokop, A., Schirmer, M., Rub, M., Lehning, M. Stocker, M. (2008): A comparison of measurement methods: terrestrial laser scanning, tachymetry and snow probing for the determination of the spatial snow-depth distribution on slopes.) Here you see laser scanning is more accurate than manual snow probing and your employer has even 2 laser scanners at least from what I know. But of course it is not mandatory, but at least you should cite this paper from your institution to have knowledge about the different accuracy standards.

Specific comments: Abstract Line 15: Delete sentence: "Such systems have the advantage that they are comparatively cost-effective and can be applied very flexibly to cover otherwise inaccessible terrain". Line 24,25: remove flexible and cost effective and investigating the worlds cryosphere Introduction: Line 24: the foot print size is not much of an issue with laser scanning anymore, about 25 per 25 cm footprint size in 1000 m distance to the scanner with very low incident angle so I would erase the sentence "TLS-accuracies suffer from acute illumination angles, resulting in unfavor25 able laser footprints, in particular within flat areas". Page 4 line 19: avalanches Test sites and data acquisition: please include here or in table 2 and 3 the following parameters, air temp., air pressure, wind speed, all at flying altitude, cloudiness, and duration of measurement/flying campaign as well as actual battery durance per flight/time, how many batteries did you use? Page 8,9. When you talk about the reference measurement using avalanche probes and GNSS please cite here Prokop et al. 2008 (Prokop, A., Schirmer, M., Rub, M., Lehning, M. Stocker, M. (2008): A comparison of measurement methods: terrestrial laser scanning, tachymetry and snow probing for the determination of the spatial snow-depth distribution on slopes) and discuss shortly the accuracy to be expected. Braemabuehl: mountain top page 12: You do here an analysis of the HS dependent on aspect. I would delete that totally as well as figure 7. It is a known fact that south facing slopes have usually lower HS if there isn't significant snow drifting involved. In my opinion this analysis has nothing to do with the actual topic of the mapping process of HS using an UAS, so I would skip that part totally. Please adapt the discussion and conclusion section to the arguments I pointed out in

the general comments section. Figure 2. Are you sure the scale bars are correct. It seems that the scale bar is the same even though the size of the images is different.

---

## Referee Comment (RC3) · Anonymous Referee #3 · 30 Mar 2016

The paper "Mapping snow depth in alpine terrain with unmanned aerial systems (UAS): potential and limitations" by Y. Buhler et al evaluates the ability and accuracy of UASs to estimate snow depth in alpine terrain. This is part of a small, but rapidly growing body, of literature that has begun to test the ability of small UASs to estimate snow depth at high spatial resolutions. This paper contributes a unique perspective by considering the accuracy of a multirotor platform in an alpine setting. The methods employed are solid and the results presented show great promise (RMSE <15 cm over grass surfaces and <30 cm over taller vegetation) as an alternative to laser scanning (airborne or terrestrial) in non-vegetated areas. I would recommend publication of these results in The Cryosphere but the manuscript does overstate the significance of this technique and the implications from this study.

I broadly agree with the comments of the other two reviews by M Nolan and A Prokop.

The evaluation of the method to estimate snow depth is solid but the wrapping text needs work. The authors are proponent of using a multirotor for this work and I do not see why this bias is so strong without a direct comparison with a fixed wing platform. The authors clearly pushed their system beyond the manufacturer recommendations (Table 1 max wind speed 12- 15 ms-1 yet they report good results in wind speeds of 20 ms-1) so just comparing manufacturer specs is an insufficient test. It would be sufficient for publication to present the results achieve with this specific platform without overstepping and making broad comments on multirotor vs. fixed wing platforms.

The writing could use some work. Many sentences are awkward or unclear to me and need rewriting (see the specific comments for a non-exhaustive list). There is inconsistent writing tense that, once corrected, will make the manuscript easier to read.

Specific Comments:

Title: while potential and limitations are in the discussion the majority of the paper deals with producing and assessing the accuracy of the snow depth maps. Perhaps the "potential and limitations" could be dropped to simplify the title

Page 2 Line 1: "spatiotemporal distribution, and variability of snow depth (HS" -> "spatiotemporal snow depth (HS) distribution" . . . may be more clear

Page 2 Line 10: The Nolan review does bring up a fair point regarding making statements/ comparing the economics of this system to manned platforms (or even other UASs). Without doing a full economic analysis these statements are merely speculative. As well, regulations affecting UAS (which are rapidly changing and vary by nation) and aircraft operations s may play a larger role in determining the application of this method than simply comparing the ticket price of equipment. With all of these competing factors, which are beyond the scope of this paper or journal, perhaps it may be more appropriate and straightforward to limit this paper an assessment of the capabilities of the method.

Page 2 Line 13: " an unmanned aerial system (UAS)" as you only used one platform-this was not an intercomparison.

Page 2 Line 19-20: "monitor the ablation"-> "monitor the snow ablation". What about at the second site? I would recommend you mention snow depth was estimated once here to keep the text balanced.

Page 2 Line 23-24 and throughout text: "better than" -> "less than"

Page 2 Line 24-26: awkward ending to this sentence. Please rewrite.

Page 3 Paragraph 1: This paragraph is a list separated by semicolons without any sort of closing to wrap up these points. Rewrite without using semicolons as it is rather awkward.

Page 3 Line 13: Remote sensing is a field of study with many different tools not a tool itself like UAV SfM. Rewrite.

Page 3 Line 15- 17: perhaps put your definitions of snow depth into a methods section.

Page 3 Line 21-23: Unnecessary sentence.

Page 3 paragraph 2 and 3: After suggested edits merge these two paragraphs.

Page 4 Line 5-6: "were not feasible to most applications" –awkward

Page 4 Line 11: "Throughout the last years," -> "Recently,"

Page 4 Line 11-16: replace semicolons with commas.

Page 4 Line 20. Replace colon with period.

Page 4 Line 20-21: As you are likely already aware de Michele 2015 in TCD is now de Michele 2016 in TC. Other recent examples can also be found in TCD (Harder et al., 2016 and Marti et al., 2016)

Page 4 Line 23-27: rewrite to avoid faulty parallelism. "implementing sensors capable

of measuring at e.g. near infrared wavelengths" is unclear

Page 5: Why are sections 2.1 and 2.2 distinct from each other?

Page 5 Line 10: can you define "high positional accuracy". How accurate is the positioning? Is this standard GPS accuracy ie +- 5m?

Page 5 Line 17-19: Can you be more explicit on the color bands with and without filters? A table would be valuable to quickly compare the EM spectrum being sampled in the various configurations.

Page 6 Line 1-20: perhaps a new section along the lines of "UAS deployment"

Page 6 Line 11:" important key" redundant. Pick one

Page 6 Line 12: delete "feasible". Redundant

Page 6 Line 13-15: the capabilities of camera by themselves do not enable generation of highly accurate DSMs. Other factors such as overlap are critical. Rewrite to clarify what you are trying to say.

Page 6 Line 19: not simply limited by weight. Also limited by space and power. In case of Ebee specifically, cameras are primarily limited by what the manufacturer offers as only Sensefly sensors can be used in the ebee.

Page 6 Line 19-22: I disagree with this simplification. The octocopter may be easily transportable but with an effective flight time of <10 minutes the operator needs to be in or directly adjacent to the area of interest. While a larger system may not be able to be transported as near to the area of interest as a multirotor it can travel further to overcome such a disadvantage. It may necessary to emphasize that the best platform for the job will be site specific.

Page 6 Line 23: Speculation. Will be site specific.

Page 6 Line 9-11: Tense is inconsistent

Page 6 Line 13-20: What parameters were used in this study? Was the accuracy of the estimated snow depths sensitive to these parameters? Was this tested?

Page 6: Point cloud generation is discussed but how are the DSMs and orthomosaics generated. This needs to be added.

Page 6 line 23 and elsewhere: change "well-accessible" to "easily accessible" or something less awkward.

Page 7 line 5: delete "quite"

Page 7 line 9: "usually not exposed" -> "not usually exposed"

Page 7 Line 9-11: how were slope angles estimated? From the DSM?

Page 7 Line 13-15: How was this overlap determined to be optimal? Was this determined through trial and error? Was this a recommendation? How do you determined DSM quality? Did you test quality versus time? Justify the selection of this overlap more clearly.

Page 7 Line 18-20: Were multiple batteries switched out during each image acquisition period or was acquisition limited to what could be acquired off a single battery. Switching out batteries greatly extends the duration of any proposed missions and this information will help potential users evaluate your experience.

Sect. 3.1 and 3.2: please include the size of the areas mapped at each site.

Page 9 Line 17: Why was NIR selected at this site and not at Tschuggen? Does this change the accuracy results? Was there a test of the different wavelengths at a common site and time to see if this would influence the accuracy results?

Page 10 Line 3: delete "e.g."

Page 10 Line 5: spelling "referene"

Page 10 Line 6: "are resulting" -> "results"

Page 11 Line 4: Do you have confidence that you were actually able estimate a mean snow depth of 1cm? Granted that this is an areal average of variable snow depth but this is a lot less than any of your estimated geolocation or snow depth errors.

Page 11 Line 18: "is an average systematic underestimation of HS by 0.2 m" is this the same as bias? Perhaps it would be good to use terms common to other papers on this topic (ie Harder et al 2016)

Page 11 Line 26-29: Are these values an average of the errors for all respective snow depth in each class for all flights? This is unclear. What is mean shift? Same as bias? Clarify/keep your terminology consistent.

Page 12 Line 3: "RMSE of $\sigma$ is 0.04 m" based on all flights? Clarify please.

Page 12 Line 13: Would it be possible to add a legend/color bar to the animation to more easily interpret the snow depths.

Page 12 Line 28: I fail to see the value of including the correlation coefficient. The RMSE is sufficient while the R2 (due to the small RMSE and large range in snow depths) will give a deceptively good value.

Page 13 Line 1: can make text more concise if you refer to this as bias (if that is what it is).

Page 13 Line 3-4: This sentence is unclear to me as to what you are comparing.

Page 13 Line 20-21: delete "which is the appropriate starting/landing procedure we apply in alpine terrain". Redundant.

Page 13 Line 21-23: Perhaps. But this is platform and site specific so such a strong universal statement is rather speculative.

Page 13 Line23-24: Did you actually fly in -30C or is this also speculative?

Page 14 Line 6-9: Without an actual comparison this is also speculation. Flying conditions will be site specific and fixed wing platforms have vastly different capabilities negating any universal conclusions.

Page 14 Line 13: delete "However,".

Page 14 Line 18: DSM instead of "DEM"?

Page 14 Line 23: delete "However,".

Page 15 Line 6-12: You used NIR and no-NIR imagery in this study already. Can you make any comments on this topic already?

Section 5.3: Coregistration is important but this section could be removed as the different methods were not compared as far as I can see and doesn't directly contribute to the results of the paper.

Page 17 Line 13: add recent papers as previously mentioned.

Page 17 Line 15-16: Maximum altitudes of UAV's is generally quite low due to regulations (which will of course vary by country) so this is likely unfeasible.

Page 18 Line 9-26: I agree with the M Nolan review that this should be moved to the discussion.

Page 19 Line 1-3: Rewrite final sentence as it is unclear.

Figure 5: in caption, do the R2 values refer to HS measurements? Clarify. Remove shading from points on plots

Figure 7: If you do keep this section (re: Prokop review) clarify what the bars and line represent (unclear which is mean snow depth and which is standard deviation). Also make bars a solid color.

Figure 8: what is the line in the HS measurement plot below the 1:1 line? Remove or explain.

Rearrange order of figures to reflect the order they are referred to in the text.

Any mention of "significance" should be removed unless backed up with statistical tests.

---

## Author Comment (AC4) · 6 Apr 2016

Dear Referee

Thank you for your very late but valuable review. In our opinion we have already answered a lot of your questions with the answers to the review of Matt Nolan and Alexander Prokop. Therefore we limit here the answers to the questions going beyond the points of the two other reviewers.

The additional point you bring up in the general comments is that we should not make statements on fixed-wing UAS as we did not use them in this study. That is true but we have quite some experience at our institute with fixed-wing UAS and did fly (and crash) them around Davos (CH) and Innsbruck (AT). We have flight experience with a SensFly eBee, a Trimble UX5 as well as self constructed devices. However, in the paper we

will mark all broader statements on fixed-wing UAS with "from our experience". In our opinion, the statements we make are well enough supported by our experience in alpine terrain and are valuable hints for readers, therefore we want to keep these statements.

Specific comments:

In our opinion this investigation reveals a lot on potential and limitations of UAS for snow depth mapping in alpine terrain. We also discuss this point. Therefore we want to keep it in the title, if the editor agrees.

P2L1: changed as suggested

P2L10: please see answers to M. Nolan and A. Prokop

P2L13: changed as suggested

P2L19-20: As we have three DSM acquired during different dates in winter at Tschuggen, we can monitor ablation processes. We cannot do that at Brämabühl as we only have one DSM acquired during winter.

P2L23-24: changed as suggested

P224-26: changed

P3P1: Changed to individual sentences

P3L13: Changed to "Remote sensing is useful to monitor"

P3L15-17: We thought a lot about the best position for this definition. As it is essential for the entire paper, we decided to bring it at this place, early in the paper.

P3L21-23: In our opinion this sentence makes sense here as it describes a recent TLS application where snow depth is the key variable.

P3: We think the text is better structured keeping the two paragraphs

P4L5-6: changed to "were insufficient for most applications "

P4L11: changed as suggested

P4L11-16: changed as suggested

P4L20: changed as suggested

P4L20-21: We update and include the recently published papers

P4L23-27: rewritten to "De Michele et al. (2016) conclude, that UAS-based HS mapping holds great potential, but that further studies are required especially with regard to multi-temporal mapping, to sensors capable of measuring in near infrared bands or to the mapping of different snow cover conditions (new snow, wet snow, ice crusts etc.)."

P5: we merge 2.1 and 2.2 as suggested

P5L10: change to "of better than 2.5 m (personal communication from Ascending Technologies)"

P5L17-19: We list the filter thresholds we have available. As we do not further use the different filters in this study, this information is sufficient in our opinion.

P6L1-20: We do not understand, where you suggest putting the section break.

P6L11: changed as suggested

P6L12: changed to "good"

P6L13-15: changed to: The radiometric and spatial resolution of the Sony NEX-7 camera enable the generation of highly accurate digital surface model (DSM)."

P6L19: changed to "due to limited carrying capacity, space and battery power."

P6L19-22: We add "from our experience" and change not appropriate for high mountain areas into "difficult to fly in high mountain areas"

P6L23: This is not a speculation but a feedback we get from nearly all colleagues flying

fixed-wing UAS in mountains. This is clearly the point causing most trouble applying fixed-wing UAS in alpine terrain. And should therefore stay in our opinion.

P6L9-11: we do not find an inconsistent tense here

P6L13-20: We describe the data acquisition parameters in the chapters 3.1 and 3.2. We did not perform a sensitivity study of the DESM quality to the parameters yet but we are planning such a study for this winter.

P6: We add "using dense point cloud generation with the default parameters"

P6L23: changed as suggested

P7L5: changed as suggested

P7L9: changed as suggested

P7L9-11: yes from the summer DSM resampled to 1 m

P7L13-15: This overlap was chosen based on discussions with different colleagues and is based on our own experience. We are planning to investigate this question in more detail in a follow on study. To make this clear we write "From our experience".

P7L18-20: we add: "The Tschuggen test site can now be covered with one battery." And "To cover the Brämabühl test site we need four batteries".

Sect 3.1 & 3.2: we already list the areas in the tables 2 & 3.

P9L17: We are currently investigating the benefit of NIR compared to RGB. This was not yet investigated for this study.

P10L3: there could be other causes why a slope is not accessible. Therefore we want to keep the e.g.

P10L5: changed as suggested

P10L6: changed as suggested

P11L4: This is the mean snow depth averaged over the entire test site. There are only a few spots with remaining snow cover (Fig. 4)

P11L18: changed to "average underestimation of HS by 0.2m". In our opinion this formulation is more precise as bias.

P11L26-29: This are the RMSE values per class (as it is written). We change mean shift to bias as suggested and add "for all three flight dates".

P12L28: The correlation between the reference HS measurements and the photogrammetrically measured HS is in our opinion a useful estimation of the mapping quality, as many readers will be used to correlation as a measure of quality. As the investigated test sites are typical for alpine catchments, want to keep these values. They depict that no drift of error occurs at very high or low HS values.

P13L1: In our opinion our description here is easier to understand and more precise

P13L3-4: we change mean deviation to bias. WE compare to the stddev within a reference plot as we write.

P13L20-21: We want to keep this information as we think it is important for the readers.

P13L21-23: we add "Based on our experience" to make this clear

P13L23-24: We did flights were we had air temperatures of -25 ° C, -30° can occur in the early mornings in Davos. We faced problems with cold batteries several times.

P14L6-9: we add "Our experience shows that"

P14L13: changed as suggested

P14L18: changed as suggested

P14L23: changed as suggested

P15L6-12: We are investigating the difference between NIR and RGB this and next winter. We do not have reliable quantitative results yet to publish them in this paper.

S5.3: Coregistration is an absolutely crucial point for photogrammetric HS mapping. Therefore we want to keep this part.

P17L13: changed as suggested

P17L15-16: In Switzerland you are allowed to fly at an altitude of 500 m above ground, so it is feasible. We cannot mention all the different regulations around the world here.

P18L9-26: Moved to the discussion as suggested

P19L1-3: rewritten to "We expect that UAS will get more and more important for mapping applications also high alpine terrain and that this methodology will change the frequency and quality of geodata acquisition fundamentally."

Fig5: we add "for the HS values" to clarify.

Fig7: we add "(bars) and (line) to clarify"

Fig8: We remove this trend line as it is too close to the 1:1 line

---

## Author Response (AR1)

Dear Matt Nolan,

We thank you for your positive feedback and the constructive comments.

"UASs enable fast, flexible, repeatable and detailed analysis of the spatial distribution of mountain snow cover". This sentence from the paper describes the essential findings of the work, though I would add "over several hectare areas" to improve the accuracy of that description. Towards these ends, the authors have conducted sound scientific experiments that are well supported and described, and I believe their work deserves to be published. The only scientific analysis I found lacking was an analysis of the repeatability of their system – measuring the same location twice on the same day (or a snow-free road on two different days) and seeing how close the measurements are to each other; that is, determining the noise level of their system, and it seems they have data in hand to do this.

The authors are clearly strong supporters of UAS technology, and I applaud and en- courage their efforts to push the boundaries of this technology towards such an impor- tant scientific subject. However, the paper reaches well beyond the scope of its scien- tific findings to make claims about the implications or justifications of this work without support for those claims. I found two categories of such claims. First are claims that UAS are somehow more cost effective to use than manned aircraft. Though I readily admit my bias, as a scientist on a budget I would not be using a manned aircraft to measure snow pack photogrammetrically if I believed this to be true. These claims need to either be removed or validated through an actual economic analysis, and this analysis needs to at least encompass variables such as region of the world, full costs for manpower, and areal coverage. For example, I can map 100 km2 at 10 cm GSD in an hour in my manned aircraft and I can do so over steep, dangerous terrain without risk being caught in an avalanche, for a total of perhaps 4-5 man-hours of field effort. By comparison, the UAS work in this paper failed to demonstrated that it could map more than 1 km2 in a day's work for several people – though its direct costs may be much less, how much salary time would it take a 2-3 man team to map 100 km2? Perhaps there are economics that I don't understand and I am happy to be educated, but in any case these statements require justification before manned aircraft can be summarily dismissed in favor of UAS due to cost. This leads to the second category of unsupported claims regarding future use of UAS for the purpose of wide-area mapping. The conclusions, for example, list 8 future uses of UASs, only one of which the authors have shown any support for within the paper. For example, claims that a UAS can make "precise water resource predictions for hydropower and flood warning in alpine catchments" – that is, that they can map 100s of km2 – have no support in the paper, and indeed the paper admits several times that the limited flight times of 10-20 minutes are a major hindrance to their research in even small areas. As another example, stay- ing in line of sight of the UAS means that the pilots must travel essentially through the dangerous avalanche terrain they claim their UAS can measure. If the authors want to assert these uses, then more validation and description is required that their system is capable of it. I'm enthusiastic about the potential uses for this technology, but I don't see that the actual uses are highlighted here.

Thus overall I think the paper would be substantially improved by changing the wrapper placed around their work and rewriting it to focus on the useful results they found and their true significance – they have shown that they can measure several hectare areas in a variety of terrain types at very high spatial resolution and very good accuracy and this will benefit many types of studies that are currently hampered by the lack of such measurements. There are plenty of such applications, no need for touting these as a replacement for manned aircraft in those many applications where manned aircraft are much more cost effective (like large area mapping) and much safer. The text could use a bit of cleanup but is overall well written and the science seems well done, supported, and verifiable, though as stated earlier a repeatability spec would improve it further.

We do not at all intend to talk down the value of manned aircraft for snow depth mapping over large areas. We are ourselves strong supporters of this method (e.g. Bühler et al. 2015). However, to cover small areas, I am positive that UAS are more economic than airplanes in most countries around the world. The situation you have in Alaska is very special. I was really jealous to see all the nice airplanes in the gardens outside Anchorage. This is really a dream come true for every remote sensing guy. However, in Switzerland and most European countries, it is quite time-consuming and costly to get an airplane and acquire the necessary flight permissions. This is additionally hampered by very restrictive ATC-regulations for small airplanes in Central European airspace and the use of alpine airports. Repetition flights are nearly impossible to do, we collected some quotations on that. On the other hand, the regulations to fly UAS are different for every country and sometimes even for different states and they are changing quickly. Therefore it is impossible to list all the different regulations. We confine us to give a brief description of the most important regulations for Switzerland in the paper. However, as you suggest, we amended the proposed addition "over several hectare areas" to the sentence in the conclusions to make this clearer. And you are right, compared to North America, the Swiss Alps have a very dense infrastructure (roads, railways and huts that serve spaghetti and fondue), which makes it much easier to deploy UAS, because you can reach most spots of interest quiet easily. On the other hand it is very difficult and expensive to get an airplane with flight permission on short notice.

We will include an assessment of the repeatability by analyzing the snow free road at the Tschuggen test site at all 4 flight dates. Thank you for this valuable suggestion. We add the following paragraph to chapter 4.1: "To assess the repeatability of the UAS HS mapping, we analyze the altitude deviation of the different DSM for 10550 grid cells on the snow-free road. The calculated RMSE values compared to the summer DSM (28 September) are 0.093 m (11 March), 0.052 m (24 April) and 0.045 m (12 May). This indicates that the noise of the method is smaller than 0.1 m."

Specific comments:

Abstract Line 1: Not really a topic sentence. Best to get as much of the who, what, where, why, and when out in the first sentence, but this is personal preference.

P1L1: The first sentence intends to give a broad picture why snow depth information is important. We therefore want to keep this sentence.

Line 2: No need for "(HS)" as you don't use it again within the Abstract

P1L2: We use the abbreviation HS for snow depth consequently through the paper, therefore we would like to already include it in the abstract.

Line 3: "Nowadays" is an odd word here

P1L3: We change Nowadays into Currently

Line 6: This sentence is not quite accurate or meaningful, as 'dense' is not defined well enough to evaluate it. A dense enough network could be devised for any locale, the question is really whether it is feasible to implement.

P1L6: We change the sentence to "even a dense measurement network like in Switzerland with more than one measurement station per 10 km$^2$ in average "

Line 10. The implication by saying 'costly' is that UAS are cheaper. Remove, or support in the paper.

P1L10: We add "in most countries" to consider the special situation in Alaska.

Line 15. Again, either provide an analysis in the paper that UAS are "comparatively cost effective" or remove the statement. Similarly about the next part of the sentence for use in "otherwise inaccessible terrain" as this was not supported in the paper as all the sites used were easily accessible, and the paper actually recognizes this as a limitation.

P1L15: We change otherwise inaccessible not accessible from the ground. Not all parts of the Brämabühl test site were accessible. The steep north-facing slope is very prone to avalanche danger and cannot be accessed during most days in winter.

Line 21. RMSE of "snow depth values"? Do you mean residuals between the measurement types? Or a mean snow depth? Or?

P1L21: We added "compared to manual snow-depth measurements" to clarify this

Line 24. Again, remove cost effective or justify, and clean up the end of the sentence a bit.

P1L24: We change the sentence as following "This new
measurement technology opens the door for efficient, flexible, repeatable and cost-effective snow depth monitoring over areas of several hectares for various applications."

Introduction I believe in this section some clear mention should be made of the true roles that UAS can play today in terms of areal coverage and contrast this with what manned aircraft can do. I use both, but I only use a UAV when I'm already on the ground somewhere. This is the place for an economic justification for the use of UAVs over manned aircraft, if there is one. Flying a manned aircraft to a remote location to drop off a team to use a UAV in a tiny area makes little scientific sense for most applications and costs more. But if you have a road or trail system through a mountain range with huts that serve spaghetti every 5 miles and you have no budget at all then using a UAV to map small areas nearby may make some sense economically. Or however you think about it, just be explicit about your claims. Please also see Nolan and Deslauriers 2015 currently in Cryosphere Discussions, where we map snow depth over the tallest and most remote peaks in the US Arctic using a manned aircraft. Here we show that we can truly map avalanche danger, cornice development, gully filling, etc, not as some future possibility but as true examples of our current capabilities. While we did not discuss economics much there, the ability to map snow depth on a big chunk of a mountain range located 350 miles away in a single flight is something that UAS will never be able to do at any cost, and this is worth bearing in mind in this paper, especially since UAS are banned in most US federal lands. Here also some mention should be made of what sorts of projects that a UAS can actually do better than can be done from a manned aircraft; if there are none, this should be stated (I think there are).

Introduction:
We put the proposed discussion on UAS capabilities to the discussion section. Are you sure you are able to map avalanche danger as you write? Your impressive results in Nolan and Deslauriers (2015) show the mapping of cornices and the filling of terrain features. However I do not see that you map avalanche danger.

P2L27: We listed three different quotations of airborne LiDAR- and Photogrammetric data acquisition in Bühler et al. (2015). It is clear, if you already have a full system including airplane and pilot, that the costs are considerably reduced. However, we doubt that there are lots of users who have all this equipment available (especially in Europe).

Page 4, Line 2. Again, provide support for "cost-effective"

P4L2: please see answer above

Methods Page 5, Line 20. Near infrared is mentioned several times throughout the paper as having advantages on snow, but I found no results of this UAS work that supported this. Perhaps I missed it, so this should either be emphasized further or this discussion toned down.

P5L20: We cite two papers demonstrating the advantage of near infrared bands. However, in this study we are not able to investigate this topic in detail. Therefore we just mention the potential by citing. We are right now working on a detailed study to quantify the benefit of near infrared bands.

Page 7, Line 13. The quality setting is directly related to resolution used in the calculations: ultra high uses each pixel individually, High uses 2x2 pixels, Medium 3x3, etc. The filtering is mostly necessitated by parallax caused by motion and match point errors I believe. This doesn't need to be mentioned in the paper, just commenting.

P7L13: We think this is interesting information for readers who want to apply SfM and we would like to keep it in the paper.

Page 7, Line 25. This sentence is confusing. It says two "well accessible" sites that are "typical locations" – does this mean most sites in these mountains are easily accessible? This relates directly back to claims earlier of being able to work in inaccessible locations.

P7L25: We usually choose locations for our investigations that are well accessible. We would have a lot of other areas, which are not as well accessible. But the terrain characteristics of the chosen test sites are representative for a lot of areas. We change typical locations into "that represent typical terrain characteristics in high-alpine environment" to clarify.

Page 8, Line 14. Do you have support for this claim of being a good compromise? I think its true, but it should be supported when stated like this.

P8L14: We did several tests with different overlaps. Our experience showed that 70% is a good compromise. We add "from our experience with different overlaps we conclude that" to clarify why we get to this conclusion.

Page 8, Line 16. I don't see anywhere in the paper or tables specs on the GPS accuracy of the UAV position? It strikes me that the 'older' version may actually be better than the newer one, because if the UAV stabilization on a location, its positional accuracy may be improved simply because there the timing error is reducing (if the position uses the camera's exif data in integer seconds). Have you explored whether the old and new methods give the same results?

P8L16: We do not use the position of the cameras recorded by the UAS GNSS as the accuracy is estimated to approximately 2.5 m, which is insufficient for snow depth mapping application. We only use the UAS GPS to fly the lines defined in the flight planning software. Unfortunately the producer of our UAS gives no details on the applied GNSS sensor.

Page 8, Line 21. The word 'selected' is repeated.

P8L21: we replace selected with applied

Page 8, Line 25. How was orthoimage accuracy measured? By eye in comparison to photo-identifiable GCPs? What does the Z value mean in terms of an orthophoto?

P8L25: This is the orientation accuracy, not the accuracy of the orthoimagery. These values are calculated from the applied RPs. We change orthorectification to "orientation".

Page 9, Line 17. I'm confused about the use of the NIR imagery. From figure 3, it looks to me that the NIR shows less detail than the other. The text says NIR is 'expected' to be better – well, was it?

P9L17: See answer to P5L20. For this study we do not have simultaneously acquired NIR and RGB data but we work on a secondary study where we will have such data available.

Page 9, Line 26. I'm confused about the use and necessity of GCPs in this study. Are these being used in the bundle adjustment at all, or just for validating the results? A clear statement needs to be made about this.

P9L26: We mixed up the correct terms. What we use are reference points RPs not control points GCPs. We use these points to absolutely orientate the photogrammetric products. We change GCPs to reference points RPs throughout the paper. The RPs are used for absolute referencing of the DSM and Orthophotos into the Swiss CH1903 LV03 coordinate system. At Tschuggen, the RPs are the same for all four flight dates and are therefore used also to reference the summer and winter DSMs. At Brämabühl we choose RPs from the absolutely referenced summer orthophoto.

Page 11, Line 7. I'm confused as to what this classification is doing? Also, why set negative snow depths to zero? There is clearly snow there, so its not zero.

P11L7: We classify the area to snow and no snow. Areas not covered by snow are set to HS = 0. Negative values are not changed but due to the color bar limit, they are depicted red (HS <= 0). We change the sentence to "…. snow-covered areas have been separated from snow free patches using a simple unsupervised classification." to clarify this point.

Page 13, Line 17. Here range and pilot positioning are discussed as being limitations. There's nothing wrong with this, it is what it is. UAVs have a place, but that place is not wide area mapping as can be done from manned aircraft. As stated earlier, I think these limitations need to be discussed in the abstract and introduction, so as not to give the reader false expectations about what UAVs are capable of, but I also don't feel that limitations are something to be ashamed of, different tools for different jobs.

P13L17: In our opinion the discussion is the right place to discuss this question, as we do. However, we added your suggested changes in the abstract restricting the application of UAS to several hectares areas. As you suggest we move the potential applications from the conclusion to this discussion chapter.

Page 15, Line 6. Again, its not clear whether the research of this paper demonstrated anything regarding NIR superiority, so its not clear to me what this paragraph's purpose is.

P15L6: In this section we discuss what could potentially improve the results. As stated before we are working on a secondary study quantifying the assumed benefits. We clearly state "However, further studies have to investigate the real benefit of NIR bands for photogrammetric HS mapping in more detail. "

Page 15, Line 19. "GPSs" is used when I think "GCPs" are meant. I found these 3 options confusing and I'm not sure what any of them mean. What's the difference between "a" and "c"? Why are GCPs needed at all – why not just co-register them and ignore the realworld coordinates? There is also a better option – just use the manual probe depths for co-registrations. This is a great advantage for UAVs – you are going to be standing in your field area anyway, so you have opportunity to probe, and then just match the UAV snow depths to those manual probe measurements, at locations where vegetation is minimal. Further, this sort of registration is only required in the first place because the on-board GPS is not accurate enough to directly georeferenced the data accurately enough for this application; this should be discussed and mentioned for future development. That is, if your photo positions had < 1cm accuracy, your maps would too, and this is a possibility for slow moving UAVs even today.

P15L19: Yes this is a mistake. We change GCPs to RPs anyway (see answer P9L26). C is a combination of a and b leading to an absolutely referenced HS map with no offsets in the HS values. As we do not use the UAS GNSS measurements (they are not accurate enough), we must apply RPs, either relative or absolute. We do not understand your proposed approach of matching the HS maps to the probe measurements as we would have a large number different cells with the HS value of the probe (plus minus lets say 5 cm of error). Furthermore we would not be able to assess the achieved accuracy of the HS product if we use the probe measurements for referencing. We add the following sentences to discuss the possibility of very accurate onboard GNSS referencing: "RPs would not be necessary if a very accurate  (better than 0.05 m) GNSS system would be available directly on the UAS. First UAS products with such high-accuracy GNSS sensor are already available on the market. However a first investigation by Harder et al. (2016) indicates that the achieved orientation accuracy is not sufficient for snow depth mapping without ground reference measurements."

Page 16, Line 9. Accuracy of what?

P16L9: We add "of the HS maps"

Page 17, Line1. This is far overstated. UAVs have particular trouble with tree motion because they are such high resolution GSD (or TSD in this case. . .). From a manned aircraft, tree motion has a negligible effect on results in the 30-50 cm GSD range, and I have tons of data at 5-10 cm GSD within forests. Whether the area beneath the tree is visible depends on the tree – our black spruce are quite skinny, and our birch tree lose their leaves in winter allowing us to map the ground beneath clearly.

P17L1: Here our experience is different. We are not able to reliably map the ground around and below trees and bushes, mainly because they are moved by wind, which is nearly always present. It is possible that this effect is less distinct within data with lower GSD. Therefore we want to keep this statement. However we change impossible into "difficult"

Conclusions This section needs a rewrite, as there is a lot of Discussion mixed in with actual review of results and findings. For example, the fixed wing UAS discussion. See also my earlier comments regarding using the manual probe depths for co-registration and to eliminate the last paragraph/list describing uses that UAVs are not capable of currently (and probably never will be) or justify these claims more fully, but in any case move to Discussion. I believe a list of true potential uses for the system that the authors used to measure snow depth (that is, snow depth with tiny GSD over small areas) would be a great idea, but this should placed in the Discussion.

Conclusion:

As suggested we move the list of potential applications to the discussion chapter. In our opinion it is not necessary to map entire catchments for water resource prediction or flood warning, but it may be sufficient to map representative subsections which can be done wit UAS. Therefore we want to keep this application in the list even though this can also be done with manned airplanes, as we showed in (Bühler et al. 2015). However we add "small" to alpine catchments.

Dear Alexander Prokop

Thank you for your fast and constructive review. In our opinion we have already answered a lot of your questions with the answers to the review of Matt Nolan. Therefore we limit here the answers to the questions going beyond the points of Matt Nolan.

First I want to point out that the authors have conducted sound scientific experiments that are well supported and described, and I believe their work deserves to be pub- lished. UAS snow depth measurements will be an useful alternative to measure spatial snow depth distributions in the future and I encourage the authors efforts to push the boundaries of this technology towards such an important scientific subject as well. I also understand and would have had probably the same enthusiasm because of the great data presented, however, when M. Nolan summarizes that the sentence: "UASs enable fast, flexible, repeatable and detailed analysis of the spatial distribution of moun- tain snow cover" describes the essential findings of the work, I have a slight different opinion. UASs enable neither fast nor flexible analysis of the spatial distribution of mountain snow cover as it depends on with what method you compare it to. If you compare it to manual snow probing you are certainly right, but not in comparison to recent technologies such as laser scanning. UASs need a lot of pre-organizing work, if it comes to snow depth measurements many ground control DGPS measurements (to achieve such an accuracy), and significant post processing time. In comparison to laser scanning not that fast. With being flexible I have the biggest issues, in my opinion UASs are everything but flexible. For commercial purposes (and that counts also for scientific applications) the legal limitations in many countries are significant. Usually you need a proper license to fly above populated areas such as ski resorts and you need to get permission for every flight from the air space administration. For example it took us 1 month of paper work to have an UAS flying on Svalbard (same size and weight). And then you are only allowed to fly in line of sight, which means in practice an area about 500 per 500 m wide, which is exactly what you present in your data. Thinking further on about suitable flying weather and illumination conditions you have in harsh mountain climate conditions such as in maritime coastal mountain weather or arctic weather rather more down days than flying days (strong wind and/or snow- fall, very cold and high altitudes (low battery), night, very flat light, etc.)! You present data from 6 measurement days with perfect weather conditions (I assume), therefore it would be interesting what wind, air-temperature as well as air-pressure and cloudi- ness occurred while completing the measurements. I have seen many researchers UAS crashing due to unfavorable conditions in mountainous terrain! So you have to learn flying an UAS first, before starting to make useful measurements. Furthermore you need significant computing time and rather powerful computer equipment to post process the data and create the DEMs. Thinking about the costs, I have the same opinion as Matt, I do not think it is cheaper to ask a company to deliver a DEM of a snow surface 500 x 500m, using a UAS or a laser scanner. In fact I know 2 companies that charge the same, they just use the laser scanner or the UAS depending on the area they have to measure. If the incident angle is sufficient enough and no shaded ar- eas exist they always use the laser scanner. So I would reduce your statement to what you explain in a later step to something like this: "In particular within flat areas, where terrain sections behind convex landforms such as hills or moraines cannot be covered, UAS based digital photogrammetry is a promising option for HS mapping in alpine ter- rain." That hits the application in reality much better in my opinion. That leads me to the applications you have in mind: "precise water resource prediction for hydropower and flood warning in alpine catchments (Jonas et al., 2009)Âz ˙ I have the same opinion as Matt, not enough coverage. Same counts for: "validation of snowpack and snow hydrology models (Bartelt and Lehning, 2002; Mote et al., 2003)" Snow pack models yes, but snow hydrology? I think you

did a good job to describe it for small catchments. "Survey of snow distribution in ski resorts to improve the track management (Damm et al., 2014)Âz ˙ I do not see that at all, track management in ski resorts is made by GPS measurements in real time from snow groomers, which is quiet sufficient, the worker knows immediately how much snow is underneath him (his snow groomer), so I do not see ski resort employees additionally flying around with a UAS (above people?) need- ing to post process the data, etc. All other applications have also been satisfactorily completed by alternative methods in the same or better accuracy, which are more flexi- ble than UAS, so why using now an UAS (laser scanners scan up to 5000 m nowadays in very fast operation speed, the measurement is basically done in 30 min. all in all) . For sure you can use an UAS for those applications, but there is no improvement to existing methods except for the already mentioned one.

You bring up a very important point, the one of legal regulations. It is true that UAS are a hot topic in the press right now and every country, or even every state and community, brings up its own regulations. And it is true, if the regulations are strict, UAS are not flexible anymore, as it seems to be the case for Svalbard. However, to make this point more clear we add the following section at the end of the introduction and move up the part from chapter 2.2:

"The regulations for flying UAS vary a lot from country to country or even between different states or communities. If it is necessary to get a flight certification / permission a long time before data acquisition, this limits the applicability and flexibility of this technology considerably. The regulations in Switzerland are quite user-friendly and are easy to fulfill as long as the UAS is within line of sight, no special permissions are necessary except you want to fly over crowds (more than several dozens of people within short distance of less than 100 m) or close to airports (Swiss regulations: http://www.bazl.admin.ch). However before applying UAS, the local regulations have to be checked carefully."

Furthermore we add "if the national and regional regulations permit the application of UAS" to the last sentence of the abstract.

We do not understand why UAS should need more pre-organizing work than laser scanning. Also with laser scanning you need reference points and reference measurements. And, if you want to scan areas with different expositions (this is usually interesting for snow depth investigations) you need more than one scanning position, rising the effort in time and costs considerably. Additionally, the entire TLS equipment, plus the power unit required for self-sufficient operation of a TLS in the field, are typically much more bulky and heavy than the UAS-equipment. If we reference the winter DSM onto the summer DSM, as we do at the test site Braemabühl, we do not need reference points at all and are very fast in data acquisition (flight time for Tschuggen ca. 5 minutes, Braemabühl ca. 15 minutes). We will add this information and information on the weather conditions as you suggest.

In our opinion UAS is not "better" than laser scanning but it is a valuable alternative /

complementary technique, as we state at several locations in the paper. From our experience, UAS is definitely cheaper than laser scanning to cover smaller areas with different expositions where you would need more than one scanning position to cover the entire area. Such cases occur very often in alpine terrain if you want to cover more than one mountain flank. So we get more and more requests from our institute to cover areas that have previously been covered by laser scanning. An important problem with long-range laser scanners such as the Riegel VZ 6000 is that they are not eye safe and you have to be sure, that nobody can look directly into the scanner also not with binoculars. This is very hard to ensure at least in the Swiss Alps. Also, the UAS device itself is about a quarter the price compared to the costs of a laser scanner. We were able to acquire TLS data simultaneously to the UAS data for this study at our Austrian test site – a publication of the results is in progress. Additionally, we plan more such simultaneous data acquisition campaigns for this and next winter.

The application in ski resorts is no problem from the regulations point of view in Switzerland. The limitation of existing dGNSS systems on snow groomers is that they only know the snow depth where they drove trough but they do not know what is next to them in particular next to the ski tracks. We have already requests from ski resort to test UAS for this purpose. However, following the suggestions of you and Matt Nolan, we limit our statements (costs, flexibility, data acquisition speed etc.) to "small areas".

In our opinion we focus in this paper on the UAS results we found within this study. The outlook on potential applications, causing critics from the reviewers, is now moved to the discussion part and clearly marked as "potential applications". However, we are convinced that such an outlook is very interesting for the readers and does not reach "beyond the scope of the study ". We discussed this point with different colleagues and they all have the opinion that such an outlook belongs into the paper. Big parts of this outlook are based on discussions and requests from SLF colleagues and we think they are valuable for the readers.

Abstract Line 15: Delete sentence: "Such systems have the ad- vantage that they are comparatively cost-effective and can be applied very flexibly to cover otherwise inaccessible terrain".

P1L15: We add "compared to manual measurements"

Line 24,25: remove flexible and cost effective and investigating the worlds cryosphere Introduction:

P1L24/25: we remove "investigation the worlds cryosphere" as suggested but we want to keep ""flexible and cost-effective" but add "for small areas".

Line 24: the foot print size is not much of an issue with laser scanning anymore, about 25 per 25 cm footprint size in 1000 m distance to the scanner with very low incident angle so I would erase the sentence "TLS-accuracies suffer from acute illumination angles, resulting in

unfavorable laser footprints, in particular within flat areas".

P2L24: this is still a big problem for a big part of our applications, as the SLF laser scanning experts report. Therefore we want to keep this sentence.

Page 4 line 19: avalanches Test sites and data acquisition: please include here or in table 2 and 3 the following parameters, air temp., air pressure, wind speed, all at flying altitude, cloudiness, and duration of measurement/flying campaign as well as actual battery durance per flight/time, how many batteries did you use?

We add the requested information on flight time, batteries used and weather conditions within the description of the test sites and data acquisition. We don't list air pressure because we do not hink it is of interest here.

Page 8,9. When you talk about the reference measure- ment using avalanche probes and GNSS please cite here Prokop et al. 2008 (Prokop, A., Schirmer, M., Rub, M., Lehning, M. Stocker, M. (2008): A comparison of mea- surement methods: terrestrial laser scanning, tachymetry and snow probing for the determination of the spatial snow-depth distribution on slopes) and discuss shortly the accuracy to be expected.

We add the suggested citiation Prokop et al. 2008 and discuss quickly the expected errors from manual measurements.

Braemabuehl: mountain top page 12: You do here an anal- ysis of the HS dependent on aspect. I would delete that totally as well as figure 7. It is a known fact that south facing slopes have usually lower HS if there isn't significant snow drifting involved. In my opinion this analysis has nothing to do with the actual topic of the mapping process of HS using an UAS, so I would skip that part totally. Please adapt the discussion and conclusion section to the arguments I pointed out in the general comments section.

Braemabühl: In our opinion such an analysis of snow depth distribution along different expositions is of value for the readers as it is a straightforward application of the UAS datasets. We perform this analysis at the exposed mountain top test-site as we expect a large influence of wind drift. Therefore we want to keep this analysis.

Figure 2. Are you sure the scale bars are correct. It seems that the scale bar is the same even though the size of the images is different.

Figure 2: The scale bars are correct. We covered slightly less area within the first data acquisition that is why the ortho image of March 11 looks a bit different.

Dear Referee

Thank you for your very late but valuable review. In our opinion we have already answered a lot of your questions with the answers to the review of Matt Nolan and Alexander Prokop. Therefore we limit here the answers to the questions going beyond the points of the two other reviewers.

The paper "Mapping snow depth in alpine terrain with unmanned aerial systems (UAS): potential and limitations" by Y. Buhler et al evaluates the ability and accuracy of UASs to estimate snow depth in alpine terrain. This is part of a small, but rapidly growing body, of literature that has begun to test the ability of small UASs to estimate snow depth at high spatial resolutions. This paper contributes a unique perspective by considering the accuracy of a multirotor platform in an alpine setting. The methods employed are solid and the results presented show great promise (RMSE <15 cm over grass surfaces and <30 cm over taller vegetation) as an alternative to laser scanning (airborne or terrestrial) in non-vegetated areas. I would recommend publication of these results in The Cryosphere but the manuscript does overstate the significance of this technique and the implications from this study.

I broadly agree with the comments of the other two reviews by M Nolan and A Prokop. The evaluation of the method to estimate snow depth is solid but the wrapping text needs work. The authors are proponent of using a multirotor for this work and I do not see why this bias is so strong without a direct comparison with a fixed wing platform. The authors clearly pushed their system beyond the manufacturer recommendations (Table 1 max wind speed 12- 15 ms-1 yet they report good results in wind speeds of 20 ms-1) so just comparing manufacturer specs is an insufficient test. It would be sufficient for publication to present the results achieve with this specific platform without overstepping and making broad comments on multirotor vs. fixed wing platforms.

The writing could use some work. Many sentences are awkward or unclear to me and need rewriting (see the specific comments for a non-exhaustive list). There is inconsistent writing tense that, once corrected, will make the manuscript easier to read.

The additional point you bring up in the general comments is that we should not make statements on fixed-wing UAS as we did not use them in this study. That is true but we have quite some experience at our institute with fixed-wing UAS and did fly (and crash) them around Davos (CH) and Innsbruck (AT). We have flight experience with a SensFly eBee, a Trimble UX5 as well as self constructed devices. However, in the paper we will mark all broader statements on fixed-wing UAS with "from our experience". In our opinion, the statements we make are well enough supported by our experience in alpine terrain and are valuable hints for readers, therefore we want to keep these statements.

Specific comments:

Title: while potential and limitations are in the discussion the majority of the paper deals with producing and assessing the accuracy of the snow depth maps. Perhaps the "potential and limitations" could be dropped to simplify the title

In our opinion this investigation reveals a lot on potential and limitations of UAS for snow depth mapping in alpine terrain. We also discuss this point. Therefore we want to keep it in the title, if the editor agrees.

Page 2 Line 1: "spatiotemporal distribution, and variability of snow depth (HS" -> "spatiotemporal snow depth (HS) distribution" . . . may be more clear

P2L1: changed as suggested

Page 2 Line 10: The Nolan review does bring up a fair point regarding making state- ments/ comparing the economics of this system to manned platforms (or even other UASs). Without doing a full economic analysis these statements are merely specula- tive. As well, regulations affecting UAS (which are rapidly changing and vary by nation) and aircraft operations s may play a larger role in determining the application of this method than simply comparing the ticket price of equipment. With all of these compet- ing factors, which are beyond the scope of this paper or journal, perhaps it may be more appropriate and straightforward to limit this paper an assessment of the capabilities of the method.

P2L10: please see answers to M. Nolan and A. Prokop

Page 2 Line 13: " an unmanned aerial system (UAS)" as you only used one platform- this was not an intercomparison.

P2L13: changed as suggested

Page 2 Line 19-20: "monitor the ablation"-> "monitor the snow ablation". What about at the second site? I would recommend you mention snow depth was estimated once here to keep the text balanced.

P2L19-20: As we have three DSM acquired during different dates in winter at Tschuggen, we can monitor ablation processes. We cannot do that at Brämabühl as we only have one DSM acquired during one winter date.

Page 2 Line 23-24 and throughout text: "better than" -> "less than"

P2L23-24: changed as suggested

Page 2 Line 24-26: awkard ending to this sentence. Please rewrite.

P224-26: we rewrote the ending

Page 3 Paragraph 1: This paragraph is a list separated by semicolons without any sort of closing to wrap up these points. Rewrite without using semicolons as it is rather awkward.

P3P1: Changed to individual sentences

Page 3 Line 13: Remote sensing is a field of study with many different tools not a tool itself like UAV SfM. Rewrite.

P3L13: Changed to "Remote sensing is useful to monitor"

Page 3 Line 15- 17: perhaps put your definitions of snow depth into a methods section.

P3L15-17: We thought a lot about the best position for this definition. As it is essential for the entire paper, we decided to bring it at this place, early in the paper.

Page 3 Line 21-23: Unnecessary sentence.

P3L21-23: In our opinion this sentence makes sense here as it describes a recent TLS application where snow depth is the key variable.

Page 3 paragraph 2 and 3: After suggested edits merge these two paragraphs.

P3: We think the text is better structured keeping the two paragraphs

Page 4 Line 5-6: "were not feasible to most applications" –awkward

P4L5-6: changed to "were insufficient for most applications "

Page 4 Line 11: "Throughout the last years," -> "Recently,"

P4L11: changed as suggested

Page 4 Line 11-16: replace semicolons with commas.

P4L11-16: changed as suggested

Page 4 Line 20. Replace colon with period.

P4L20: changed as suggested

Page 4 Line 20-21: As you are likely already aware de Michele 2015 in TCD is now de Michele 2016 in TC. Other recent examples can also be found in TCD (Harder et al., 2016 and Marti et al., 2016)

P4L20-21: We update and include the recently published papers

Page 4 Line 23-27: rewrite to avoid faulty parallelism. "implementing sensors capable of measuring at e.g. near infrared wavelengths" is unclear

P4L23-27: rewritten to "De Michele et al. (2016) conclude, that UAS-based HS mapping holds great potential, but that further studies are required especially with regard to multi-temporal mapping, to sensors capable of measuring in near infrared bands or to the mapping of different snow cover conditions (new snow, wet snow, ice crusts etc.)."

Page 5: Why are sections 2.1 and 2.2 distinct from each other?

P5: we merge 2.1 and 2.2 as suggested

Page 5 Line 10: can you define "high positional accuracy". How accurate is the posi- tioning? Is this standard GPS accuracy ie +- 5m?

P5L10: change to "of better than 2.5 m (personal communication from Ascending Technologies)"

Page 5 Line 17-19: Can you be more explicit on the color bands with and without filters? A table would be valuable to quickly compare the EM spectrum being sampled in the various configurations.

P5L17-19: We list the filter thresholds we have available. As we do not further use the different filters in this study, this information is sufficient in our opinion.

Page 6 Line 1-20: perhaps a new section along the lines of "UAS deployment"

P6L1-20: We do not understand, where you suggest putting the section break.

Page 6 Line 11:" important key" redundant. Pick one

P6L11: changed as suggested

Page 6 Line 12: delete "feasible". Redundant

P6L12: changed to "good"

Page 6 Line 13-15: the capabilities of camera by themselves do not enable generation of highly accurate DSMs. Other factors such as overlap are critical. Rewrite to clarify what you are trying to say.

P6L13-15: changed to: "The radiometric and spatial resolution of the Sony NEX-7 camera enable the generation of highly accurate digital surface model (DSM)."

Page 6 Line 19: not simply limited by weight. Also limited by space and power. In case of Ebee specifically, cameras are primarily limited by what the manufacturer offers as only Sensefly sensors can be used in the ebee.

P6L19: changed to "due to limited carrying capacity, space and battery power."

Page 6 Line 19-22: I disagree with this simplification. The octocopter may be easily transportable but with an effective flight time of <10 minutes the operator needs to be in or directly adjacent to the area of interest. While a larger system may not be able to be transported as near to the area of interest as a multirotor it can travel further to overcome such a disadvantage. It may necessary to emphasize that the best platform for the job will be site specific.

P6L19-22: We add "from our experience" and change not appropriate for high mountain areas into "difficult to fly in high mountain areas"

Page 6 Line 23: Speculation. Will be site specific.

P6L23: This is not a speculation but a feedback we get from nearly all colleagues flying fixed-wing UAS in mountains. This is clearly the point causing most trouble applying fixed-wing UAS in alpine terrain. And should therefore stay in our opinion.

Page 6 Line 9-11: Tense is inconsistent

P6L9-11: we do not find an inconsistent tense here

Page 6 Line 13-20: What parameters were used in this study? Was the accuracy of the estimated snow depths sensitive to these parameters? Was this tested?

P6L13-20: We describe the data acquisition parameters in the chapters 3.1 and 3.2. We did not perform a sensitivity study of the DSM quality to the parameters yet but we are planning such a study for this winter.

Page 6: Point cloud generation is discussed but how are the DSMs and orthomosaics generated. This needs to be added.

P6: We add "using dense point cloud generation with the PhotoScan default parameters"

Page 6 line 23 and elsewhere: change "well-accessible" to "easily accessible" or some- thing less awkward.

P6L23: changed as suggested

Page 7 line 5: delete "quite"

P7L5: changed as suggested

Page 7 line 9: "usually not exposed" -> "not usually exposed"

P7L9: changed as suggested

Page 7 Line 9-11: how were slope angles estimated? From the DSM?

P7L9-11: yes from the summer DSM resampled to 1 m

Page 7 Line 13-15: How was this overlap determined to be optimal? Was this deter- mined through trial and error? Was this a recommendation? How do you determined DSM quality? Did you test quality versus time? Justify the selection of this overlap more clearly.

P7L13-15: This overlap was chosen based on discussions with different colleagues and is based on our own experience. We are planning to investigate this question in more detail in a follow on study. To make this clear we write "From our experience".

Page 7 Line 18-20: Were multiple batteries switched out during each image acquisi- tion period or was acquisition limited to what could be acquired off a single battery. Switching out

batteries greatly extends the duration of any proposed missions and this information will help potential users evaluate your experience.

P7L18-20: we add: "The Tschuggen test site can now be covered with one battery." And "To cover the Brämabühl test site we need four batteries".

Sect. 3.1 and 3.2: please include the size of the areas mapped at each site.

Sect 3.1 & 3.2: we already list the areas in the tables 2 & 3.

Page 9 Line 17: Why was NIR selected at this site and not at Tschuggen? Does this change the accuracy results? Was there a test of the different wavelengths at a common site and time to see if this would influence the accuracy results?

P9L17: We are currently investigating the benefit of NIR compared to RGB. This was not yet investigated for this study.

Page 10 Line 3: delete "e.g."

P10L3: there could be other causes why a slope is not accessible. Therefore we want to keep the e.g.

Page 10 Line 5: spelling "referene"

P10L5: changed as suggested

Page 10 Line 6: "are resulting" -> "results"

P10L6: changed as suggested

Page 11 Line 4: Do you have confidence that you were actually able estimate a mean snow depth of 1cm? Granted that this is an areal average of variable snow depth but this is a lot less than any of your estimated geolocation or snow depth errors.

P11L4: This is the mean snow depth averaged over the entire test site. There are only a few spots with remaining snow cover (Fig. 4)

Page 11 Line 18: "is an average systematic underestimation of HS by 0.2 m" is this the same as bias? Perhaps it would be good to use terms common to other papers on this topic (ie Harder et al 2016)

P11L18: changed to "average underestimation of HS by 0.2m". In our opinion this formulation is more precise as bias.

Page 11 Line 26-29: Are these values an average of the errors for all respective snow depth in each class for all flights? This is unclear. What is mean shift? Same as bias? Clarify/keep your terminology consistent.

P11L26-29: This are the RMSE values per class (as it is written). We change mean shift to

bias as suggested and add "for all three flight dates".

Page 12 Line 3: "RMSE of σ is 0.04 m" based on all flights? Clarify please.

We add "based on all reference measurements"

Page 12 Line 13: Would it be possible to add a legend/color bar to the animation to

more easily interpret the snow depths.

We will try to add a legend tot he animation

Page 12 Line 28: I fail to see the value of including the correlation coefficient. The RMSE is sufficient while the R2 (due to the small RMSE and large range in snow depths) will give a deceptively good value.

P12L28: The correlation between the reference HS measurements and the photogrammetrically measured HS is in our opinion a useful estimation of the mapping quality, as many readers will be used to correlation as a measure of quality. As the investigated test sites are typical for alpine catchments, want to keep these values. They depict that no drift of error occurs at very high or low HS values.

Page 13 Line 1: can make text more concise if you refer to this as bias (if that is what it is).

P13L1: In our opinion our description here is easier to understand and more precise

Page 13 Line 3-4: This sentence is unclear to me as to what you are comparing.

P13L3-4: we change mean deviation to bias. We compare to the standard deviation within a reference plot as we write.

Page 13 Line 20-21: delete "which is the appropriate starting/landing procedure we apply in alpine terrain". Redundant.

P13L20-21: We want to keep this information as we think it is important for the readers.

Page 13 Line 21-23: Perhaps. But this is platform and site specific so such a strong universal statement is rather speculative.

P13L21-23: we add "Based on our experience" to make this clear

Page 13 Line23-24: Did you actually fly in -30C or is this also speculative?

P13L23-24: We did flights were we had air temperatures of -25 ° C, -30° can occur in the early mornings in Davos. We faced problems with cold batteries several times.

Page 14 Line 6-9: Without an actual comparison this is also speculation. Flying conditions

will be site specific and fixed wing platforms have vastly different capabilities negating any universal conclusions.

P14L6-9: we add "Our experience shows that"

Page 14 Line 13: delete "However,".

P14L13: changed as suggested

Page 14 Line 18: DSM instead of "DEM"?

P14L18: changed to "DSM" as suggested

Page 14 Line 23: delete "However,".

P14L23: changed as suggested

Page 15 Line 6-12: You used NIR and no-NIR imagery in this study already. Can you make any comments on this topic already?

P15L6-12: We are investigating the difference between NIR and RGB this and next winter. We do not have reliable quantitative results yet to publish them in this paper.

Section 5.3: Coregistration is important but this section could be removed as the differ- ent methods were not compared as far as I can see and doesn't directly contribute to the results of the paper.

S5.3: Coregistration is an absolutely crucial point for photogrammetric HS mapping. Therefore we want to keep this part.

Page 17 Line 13: add recent papers as previously mentioned.

P17L13: changed as suggested

Page 17 Line 15-16: Maximum altitudes of UAV's is generally quite low due to regula- tions (which will of course vary by country) so this is likely unfeasible.

P17L15-16: In Switzerland you are allowed to fly at an altitude of 500 m above ground, so it is feasible. We cannot mention all the different regulations around the world here.

Page 18 Line 9-26: I agree with the M Nolan review that this should be moved to the discussion.

P18L9-26: Moved to the discussion as suggested

Page 19 Line 1-3: Rewrite final sentence as it is unclear.

P19L1-3: rewritten to "We expect that UAS will get more and more important for mapping applications also high alpine terrain and that this methodology will change the frequency and quality of geodata acquisition fundamentally."

Figure 5: in caption, do the R2 values refer to HS measurements? Clarify. Remove shading from points on plots

Fig5: we add "for the HS values" to clarify.

Figure 7: If you do keep this section (re: Prokop review) clarify what the bars and line represent (unclear which is mean snow depth and which is standard deviation). Also make bars a solid color.

 Fig7: we add "(bars) and (line) to clarify"

Figure 8: what is the line in the HS measurement plot below the 1:1 line? Remove or explain. Rearrange order of figures to reflect the order they are referred to in the text.

Fig8: We remove this trend line as it is too close to the 1:1 line

Any mention of "significance" should be removed unless backed up with statistical tests.

Removed as suggested

[revised manuscript text omitted]

---

## Author Response (AR2)

Dear Editor

Thank you for you helpful comments. As requested we list your comments and our answers in this document.

3D Animation. No way to include a color bar?
We finally succeeded to include the color bar, we will submit the updated 3D animation with this revision

The line numbers below refer to the track-change version of the MS

HS not needed in the abstract. Also Defined twice in the main text (L32 not needed).
Delete as suggested

L49. Check tense. Past needed here I think.
Changed as suggested

L98. End sentence after fulfil.
Changed as suggested

L99. "except you want to fly". Improve wording.
Changed to „The regulations in Switzerland are quite user-friendly and are easy to fulfill. The UAS has to be within line of sight and the pilot is able to interrupt the flight at any time. Special permissions are only necessary if crowds (more than 24 people within short distance) are present within the overflown area or the area is  close to an airport (Swiss regulations: http://www.bazl.admin.ch)."

L120. I concur with rev#3. You need to recall in the text the size of the surveyed area, the table is good but not sufficient. To complement the table (in km²), could you give the size on the text in length * length (i.e., 300 m * 400 m for example).
Changed as suggested

L133. "concluding" from "his own experience" is a bit too conclusive. Downplay, i.e. use other word than "conclude"
Changed to „estimate"

L192. Write also 2015 after the winter dates
Changed as suggested

L216. Interesting addition. However I do not think this is "noise" (noise is what you describe in the shadow of the church for example). What about "systematic error"?
Changed to „error". We do not want to say systematic error as it is randomly distributed around the 0 value. It is not a bias.

[revised manuscript text omitted]